# TtBA: Two-third Bridge Approach for Decision-Based Adversarial Attack

**Feiyang Wang** [1 2]  **Xingquan Zuo** [1 2]  **Hai Huang** [1 2]  **Gang Chen** [3]

## Abstract

A key challenge in black-box adversarial attacks is the high query complexity in hard-label settings, where only the top-1 predicted label from the target deep model is accessible. In this paper, we propose a novel normal-vector-based method called *Two-third Bridge Attack* (TtBA). A innovative *bridge direction* is introduced which is a weighted combination of the current unit perturbation direction and its unit normal vector, controlled by a weight parameter $k$. We further use binary search to identify $k = k_{\text{bridge}}$, which has identical decision boundary as the current direction. Notably, we observe that $k = 2/3k_{\text{bridge}}$ yields a near-optimal perturbation direction, ensuring the stealthiness of the attack. In addition, we investigate the critical importance of local optima during the perturbation direction optimization process and propose a simple and effective approach to detect and escape such local optima. Experimental results on MNIST, FASHION-MNIST, CIFAR10, CIFAR100, and ImageNet datasets demonstrate the strong performance and scalability of our approach. Compared to state-of-the-art non-targeted and targeted attack methods, TtBA consistently delivers superior performance across most experimented datasets and deep learning models. Code is available at https://github.com/BUPTAIOC/TtBA.

## 1. Introduction

**Background and motivation**. Although deep neural networks (DNNs) have demonstrated remarkable performance

[1] School of Computer Science, Beijing University of Posts and Telecommunications, Beijing, China  [2] Key Laboratory of Trustworthy Distributed Computing and Services, Ministry of Education, Beijing, China  [3] School of Engineering and Computer Science, Victoria University of Wellington, Wellington, New Zealand. Correspondence to: Xingquan Zuo <zuoxq@bupt.edu.cn>.

*Proceedings of the 42nd International Conference on Machine Learning*, Vancouver, Canada. PMLR 267, 2025. Copyright 2025 by the author(s).

across a wide range of real-world applications, they remain significantly vulnerable to adversarial attacks (Biggio et al., 2013; Brendel et al., 2018; Park et al., 2024). Research on these attacks is pivotal in advancing the development of attack-resistant DNNs in the future (Bai et al., 2023).

Adversarial attack approaches are generally classified into three categories: *white-box* attacks (Goodfellow et al., 2015; Madry et al., 2018), *gray-box* attacks (soft-label attacks, score-based attacks) (Chen et al., 2017; Liu et al., 2019a), and *black-box* attacks (Brendel et al., 2018; Chen & Gu, 2020). White-box and gray-box attacks rely on complete or partial knowledge of the target model, such as its architecture, trainable parameters, or output probabilities, which is often impractical in real-world settings (Long et al., 2022). As a more practical alternative, black-box attacks are typically divided into *transfer-based* attacks and *decision-based* attacks. Transfer-based attacks (Feng et al., 2022; Ghosh et al., 2022; Fan et al., 2024; Wang et al., 2024; Sun et al., 2024; Park et al., 2024) train a surrogate model using the target model's training data. Adversarial examples are crafted using white-box attack methods on this surrogate model. However, the success of this approach is not guaranteed, due to unreliable transferability of adversarial examples (Reza et al., 2023).

**Significance of decision-based attacks**. Decision-based attacks (Brendel et al., 2018; Li et al., 2021; Shi et al., 2022; Chen et al., 2020; Chen & Gu, 2020; Reza et al., 2023) do not rely on specific details of the target DNN, such as training data, network structure, or output probabilities. Instead, adversarial examples are crafted using only feedback from the DNN's top-1 predicted label, which is the class with the highest confidence score assigned by the model. This makes decision-based attacks the most popular strategy in practice (Dong et al., 2019; Brunner et al., 2019) and the key focus of this paper. Decision-based attacks aim to deceive the target DNN (e.g., an image classifier) with minimal perturbation strength while adhering to a predefined query budget (Brendel et al., 2018). To improve efficiency, these attacks focus on optimizing adversarial examples by exploring *perturbation directions* and their corresponding *decision boundaries* (see Appendix A for more details).

**Normal vector-based attacks**. Many decision-based adversarial attacks, such as HSJA (Chen et al., 2020), Tangent

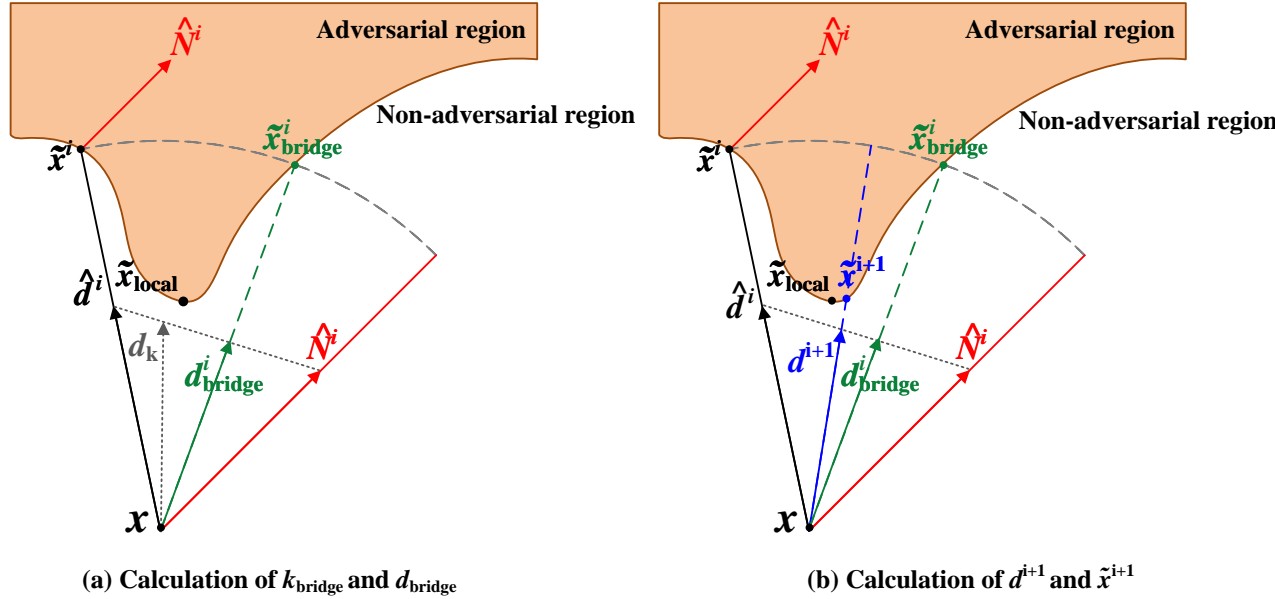

**(a) Calculation of $k_{\text{bridge}}$ and $d_{\text{bridge}}$**     **(b) Calculation of $d^{i+1}$ and $\tilde{x}^{i+1}$**

*Figure 1.* A geometric illustration of one iteration in TtBA. In (a), at the $i$-th iteration, $x$ represents the original image, $\hat{d}^i$ is the unit vector of the current perturbation direction, $\tilde{x}^i$ is the boundary point along $\hat{d}^i$, $\hat{N}^i$ is the unit normal vector at $\tilde{x}^i$, and $\tilde{x}_{\text{local}}$ denotes the local best adversarial example. We define a direction $d_k = k \cdot \hat{N}^i + (1-k) \cdot \hat{d}^i$, where $k \in (0, 1]$ is a weight parameter. Using binary search, we can identify $k = k_{\text{bridge}}^i$ such that the direction $d_{\text{bridge}}^i = k_{\text{bridge}}^i \cdot \hat{N}^i + (1 - k_{\text{bridge}}^i) \cdot \hat{d}^i$ has identical decision boundary as $\hat{d}^i$, i.e., $g(d_{\text{bridge}}^i) = g(\hat{d}^i)$. We then use $k = 2/3 k_{\text{bridge}}^i$ to generate the next direction directly: $d^{i+1} = 2/3 k_{\text{bridge}}^i \cdot \hat{N}^i + (1 - 2/3 k_{\text{bridge}}^i) \cdot \hat{d}^i$, as shown in (b). The boundary point $\tilde{x}^{i+1} = x + g(d^{i+1}) \cdot \hat{d}^{i+1}$ produces the next adversarial example.

Attack (TA) (Ma et al., 2021), GeoDA (Rahmati et al., 2020), QEBA (Li et al., 2020), and CGBA (Reza et al., 2023), exploit the *normal vector* of the decision boundary to improve attack efficiency and effectiveness. They introduce Gaussian noises to perturb a boundary point ($\tilde{x}^i$ in Figure 1) and query the model to identify which perturbations fall within the adversarial region, enabling accurate estimation of the normal vector ($\hat{N}^i$ in Figure 1) (see Appendix E for details). In recent studies (Chen et al., 2020), researchers show that the locally optimal adversarial example $\tilde{x}_{\text{local}}$ typically lies on the 2D hypersurface spanned by $\hat{d}^i$ and $\hat{N}^i$, where the normal vector $\hat{N}^i$ points directly toward $\tilde{x}_{\text{local}}$, as illustrated in Figure 1. Leveraging this geometric property, new boundary points closer to $\tilde{x}_{\text{local}}$ can be discovered as a weighted combination of $\hat{N}^i$ and the current boundary point $\tilde{x}^i$ (Chen et al., 2020; Reza et al., 2023).

The *curvatures* of decision boundaries have strong impact on perturbation optimization (Ma et al., 2021; Reza et al., 2023). In particular, targeted attacks often create narrower adversarial regions with higher decision boundary curvatures than non-targeted attacks, making perturbation optimization more challenging than non-targeted attacks (Reza et al., 2023) (see Appendix A for more details). G-TA (Ma et al., 2021) and CGBA-H (Reza et al., 2023) have been recently developed for targeted attacks on narrow adversar-

ial regions with high decision boundary curvatures. While improving query efficiency, they overlooked local optima caused by the varying geometries of decision boundaries and target models, compromising their performance across diverse DNNs (see experiment results in Section 5). Our research reveals that narrow adversarial regions, whether in targeted or non-targeted attacks, can trap perturbation optimization in local optima. To address this, we propose a simple yet effective method to detect narrow adversarial regions. An effective mechanism is further developed to escape the associated local optima and enhance performance.

**Proposed method**. We propose a novel *Two-third Bridge Attack* (TtBA) method. In this method, a new *bridge* metric, $k_{\text{bridge}}$, is introduced to guide perturbation optimization based on the decision boundary curvature.

For efficiency and simplicity, TtBA generates new perturbation directions at the $i$-th iteration ($i \geq 1$) by combining the current unit perturbation direction $\hat{d}^i$ with its unit normal vector $\hat{N}^i$ through $k\hat{N}^i + (1-k)\hat{d}^i$, where $k \in (0, 1]$ is a *weight parameter*. A geometric illustration of TtBA is shown in Figure 1-(a). In TtBA, the decision-based attack becomes simply the problem of optimizing the weight parameter $k$. We aim to find suitable $k$ to properly control the influence of the normal vector $\hat{N}^i$ on the updated direction

$d_k$. If $k$ is too large, $d_k$ moves away from the best adversarial example ($\tilde{x}_{\text{local}}$ in Figure 1), making optimization ineffective. To identify the upper bound of $k$, we introduce a threshold $k = k_{\text{bridge}}^i$ such that the decision boundary along $d_k$ equals that of the current direction $\hat{d}^i$. $k$ is then optimized within the range $(0, k_{\text{bridge}}^i]$. Geometrically, the range of perturbation directions generated by TtBA resembles an *arch bridge* over the local best adversarial example $\tilde{x}_{\text{local}}$, spanning from $\tilde{x}^i$ to $\tilde{x}_{\text{bridge}}^i$, as shown in Figure 1.

Our introduction of the new metric $k_{\text{bridge}}^i$ reveals the following interesting properties. First, the decision boundary's curvature can be directly estimated through $k_{\text{bridge}}^i$, as illustrated in Figure 2 and analyzed theoretically in Appendix B.3. For narrow adversarial regions with high decision boundary curvatures as shown in Figure 2-(c), $k_{\text{bridge}}^i$ is small. In contrast, for wide adversarial regions in Figures 2-(a) and 2-(b), $k_{\text{bridge}}^i$ is large. Second, we discover that the perturbation direction generated by $k = 2/3k_{\text{bridge}}^i$ is very close to the optimal weight in the $i$-th iteration, as shown in Figure 1-(b) and detailed in Section 4. This finding eliminates the need for searching the optimal weight extensively, thereby significantly improving query efficiency. Third, a narrow adversarial region can potentially trap the optimization process in local optima. Therefore, when $k_{\text{bridge}}^i$ is very small (e.g., less than 0.1), we increase $k$ (e.g., from $2/3k_{\text{bridge}}^i$ to $0.9k_{\text{bridge}}^i$), allowing $d^{i+1}$ to deviate significantly from $\tilde{x}_{\text{local}}$ and hence escape local optima.

In summary, our main contributions are as follows: (1) We introduce a novel metric $k_{\text{bridge}}$ to detect varied curvatures of decision boundaries, providing valuable insights into the geometrical characteristics of adversarial attacks. (2) We uncover a previously unidentified linear relationship between $k_{\text{bridge}}$ and the near-optimal perturbation direction. This insight paves the way for developing an efficient approach (TtBA) for decision-based attacks. (3) We identify and address the critical challenge of local optima in perturbation optimization, particularly in narrow adversarial regions with high decision boundary curvatures, proposing an efficient detection and escape mechanism that significantly improves attack performance. (4) Extensive experiments on 9 widely used deep models, spanning MNIST, FASHION-MNIST, CIFAR10, CIFAR100, and ImageNet, show that TtBA consistently outperforms four state-of-the-art decision-based attacks in both targeted and non-targeted settings.

## 2. Related Work

Decision-based attacks represent one of the most challenging settings for generating adversarial examples. Existing decision-based attacks can be divided into *random search attacks* (Brendel et al., 2018; Brunner et al., 2019; Cheng et al., 2019; 2020; Chen & Gu, 2020; Li et al., 2021; Maho

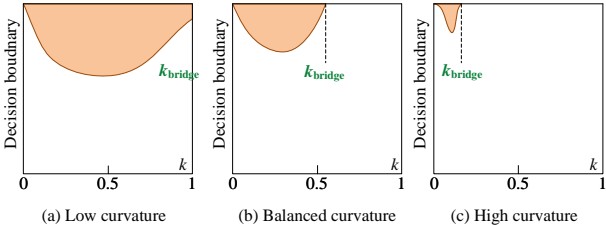

*Figure 2.* Example decision boundaries with different curvatures.

et al., 2021; Wang et al., 2022), and *normal-vector based attacks* (Chen et al., 2020; Liu et al., 2019a; Li et al., 2020; Rahmati et al., 2020; Ma et al., 2021; Reza et al., 2023).

**Random search attacks**. Random search attacks create candidate perturbations by sampling randomly, followed by optimization through binary search along the decision boundary during each iteration. For example, Boundary Attack(Brendel et al., 2018), Biased Boundary Attack (Brunner et al., 2019) and AHA (Li et al., 2021) perform random walks along the decision boundary to refine perturbation directions. SurFree (Maho et al., 2021) explores multiple directions without using normal vectors. Triangle Attack (Wang et al., 2022) utilizes a triangle-shaped perturbation structure and low-frequency spaces for efficient optimization. OPT (Cheng et al., 2019) and Sign-OPT (Cheng et al., 2020) reformulate hard-label attacks as continuous optimization problems solved through zeroth-order methods. RayS in (Chen & Gu, 2020) employs a progressive direction subdivision strategy, iteratively refining blocks of perturbation directions to enhance search efficiency.

**Normal vector-based attacks**. Normal vector-based attacks utilize normal vectors at the boundary points to guide perturbation optimization. For instance, HSJA (Chen et al., 2020) estimates normal vectors at boundary points for efficient adversarial example generation. TA (Ma et al., 2021) extends this idea by leveraging virtual hemisphere tangents to minimize perturbations. Efficient gradient estimation is achieved in qFool (Liu et al., 2019b) and GeoDA (Rahmati et al., 2020) by exploiting the observation that decision boundaries typically have low curvatures near adversarial examples. QEBA (Li et al., 2020) reduces query complexity through subspace optimization across spatial, frequency, and intrinsic dimensions. CGBA (Reza et al., 2023) introduces a novel semicircular search strategy on a 2D plane to efficiently handle geometric complexities. BounceAttack (Wan et al., 2024) improves upon HSJA by leveraging orthogonal gradient components and introducing smooth search mechanisms.

Although existing methods employ various geometric approaches for perturbation optimization, they fall short in thoroughly analyzing how boundary curvatures vary across

different models and datasets. This paper addresses these limitations by introducing novel techniques to identify narrow adversarial regions with high-curvature decision boundaries and effectively escape associated local optima, significantly enhancing attack effectiveness and efficiency.

## 3. Problem Definition

Let $x = (p_{1,1,1}, \ldots, p_{C,W,H})$, where $p_{c,w,h} \in [0, 1]$, represent a source image with shape $C \times W \times H$, where $C$, $W$, and $H$ correspond to the image's channels, width, and height, respectively. Let $y(x)$ denote the true label of $x$, and $f : x \rightarrow \{1, \ldots, K\}$ represent a $K$-class image classification model. Given a source image $x$, which is correctly classified by the model (i.e., $f(x) = y(x)$). A *decision-based black-box attacker* can only query the top-1 classification label $f(x)$ and have no access to the internal structures and parameters of the classifier $f$. The goal is to find an adversarial example $\tilde{x} = (\tilde{p}_{1,1,1}, \ldots, \tilde{p}_{C,W,H}), \tilde{p}_{c,w,h} \in [0, 1]$, such that $f(\tilde{x}) \neq y(x)$ for non-targeted attacks, or $f(\tilde{x}) = f(x_{\text{target}})$ for targeted attacks where $x_{\text{target}}$ is a given target image and $f(x_{\text{target}}) \neq y(x)$, while minimizing the perturbation strength $\|\tilde{x} - x\|_v$. The $v$ stands for the norm used to measure the perturbation strength, such as $\ell_2$ or $\ell_\infty$ (Zhou et al., 2025). We adopt the $\ell_2$ norm, following many existing studies (Chen et al., 2020; Reza et al., 2023). The problem for optimizing the adversarial example $\tilde{x}$ can be formulated as:

$$\arg \min_{\tilde{x}} \|\tilde{x} - x\|_2 \quad \text{s.t. } I(\tilde{x}) = 1, \tag{1}$$

where $I(\cdot)$ is an indicator function that determines whether the adversarial example $\tilde{x}$ is in the adversarial regions. For a non-targeted attack:

$$I(\tilde{x}) = \begin{cases} 1, & \text{if } f(\tilde{x}) \neq y(x), \\ -1, & \text{otherwise.} \end{cases} \tag{2}$$

For a targeted attack with a targeted image $x_{\text{target}}$:

$$I(\tilde{x}) = \begin{cases} 1, & \text{if } f(\tilde{x}) = f(x_{\text{target}}), \\ -1, & \text{otherwise.} \end{cases} \tag{3}$$

To ease understanding, prior works (Brendel et al., 2018; Cheng et al., 2019; Reza et al., 2023) approximate the optimization space of Equation (1) through a 2D-plane, as shown in Figure 1. They also exploit the perturbation direction (denoted by $d$) and its corresponding decision boundary (denoted by $g(d)$) to generate adversarial examples.

Let $d = (v_{1,1,1}, \ldots, v_{C,W,H}), v_{c,w,h} \in [-1, 1]$, represents a perturbation direction. An adversarial example $\tilde{x}$ can be defined as $\tilde{x} = clip(x + (\tilde{x} - x)) = clip(x + d) = clip(x + \|d\|_2 \cdot \hat{d})$, where $clip(\cdot)$ constrains each pixel to the range $[0, 1]$, and $\hat{d} = \frac{d}{\|d\|_2}$ is the unit vector obtained by normalizing $d$ using the $\ell_2$ norm. For any direction $d$,

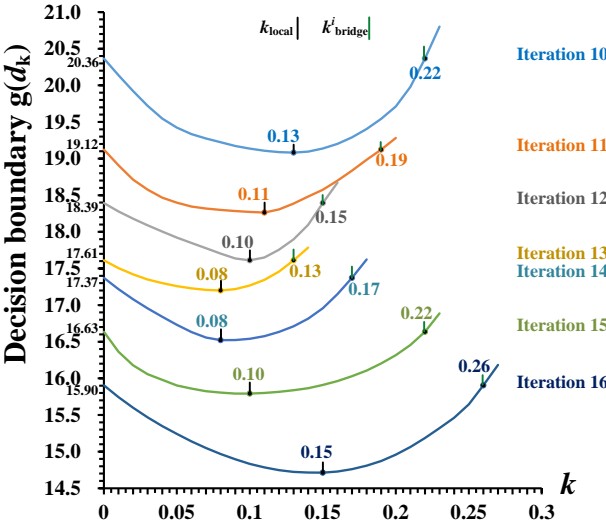

*Figure 3.* The $g(d_k) - k$ curve of decision boundary $g(d_k) = g(k \cdot \hat{N} + (1 - k) \cdot \hat{d})$ at multiple iterations of TtBA.

its decision boundary is defined as $g(d) = \min\{r > 0 : I(x + r \cdot \hat{d}) = 1\}$, with the corresponding boundary point $\tilde{x} = x + g(d) \cdot \hat{d}$. Correspondingly, the optimization problem in Equation (1) can be re-defined (Reza et al., 2023) as:

$$\arg \min_d g(d) \quad \text{s.t. } I(x + g(d) \cdot \hat{d}) = 1. \tag{4}$$

The decision boundary $g(d)$ for direction $d$ can be estimated using binary search (Reza et al., 2023), as outlined in Algorithm 2 in Appendix D.

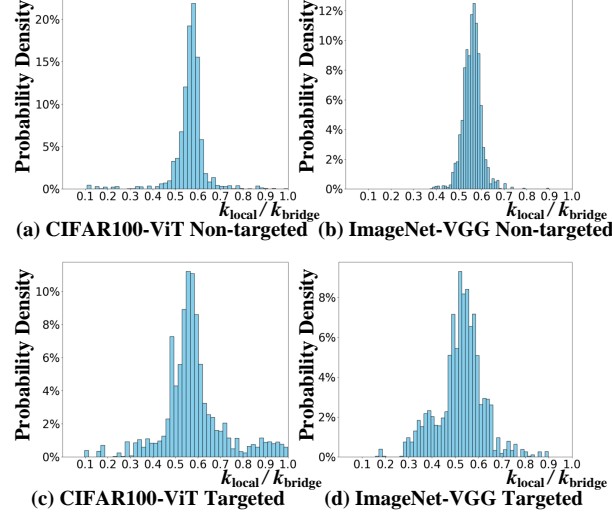

*Figure 4.* Distribution of the local best weight parameter $k_{\text{local}}$ for generating the local best adversarial example $\tilde{x}_{\text{local}}$.

**Algorithm 1** Two-third Bridge Attack

1: **Input:** Original image $x$, indicator $I(\cdot)$, budget $Q$;
2: **Output:** Adversarial example $\tilde{x}$;
3: **Initialization:** Iteration number $i \leftarrow 1$, $d^i \leftarrow$ Algorithm $4(x, I, Q)$, $\tilde{x}^i \leftarrow x + g(d^i) \cdot \hat{d}^i$, $k_{\text{bridge}}$ search tolerance $\delta \leftarrow 0.001$, local optimum threshold $\check{k} \leftarrow 0.1$, $\hat{k} \leftarrow 0.2$, bridge coefficient $\hat{b} \leftarrow 0.9$, $\check{b} \leftarrow 2/3$;
4: **repeat**
5:    $N^i \leftarrow$ Algorithm $3(x, I, \tilde{x}^i, 30\sqrt{i})$
6:    $k_{\text{left}}, k_{\text{right}} \leftarrow 0, 1$
7:    **repeat**
8:       $k_{\text{mid}} \leftarrow \frac{k_{\text{left}} + k_{\text{right}}}{2}$
9:       $d_{\text{mid}} \leftarrow k_{\text{mid}} \cdot \hat{N}^i + (1 - k_{\text{mid}}) \cdot \hat{d}^i$
10:      $\tilde{x}_{\text{mid}} \leftarrow x + g(d^i) \cdot \hat{d}_{\text{mid}}$
11:      **if** $I(\tilde{x}_{\text{mid}}) = 1$ **then**
12:         $k_{\text{left}} \leftarrow k_{\text{mid}}$
13:      **else**
14:         $k_{\text{right}} \leftarrow k_{\text{mid}}$
15:      **end if**
16:    **until** $k_{\text{right}} - k_{\text{left}} \leq \delta$
17:    $k^i_{\text{bridge}} \leftarrow k_{\text{left}}$
18:    **if** $k^i_{\text{bridge}} \geq \hat{k}$ **then**
19:      $k^{i+1} \leftarrow \check{b} \cdot k^i_{\text{bridge}}$
20:    **else if** $k^i_{\text{bridge}} \leq \check{k}$ **then**
21:      $k^{i+1} \leftarrow \hat{b} \cdot k^i_{\text{bridge}}$
22:    **else**
23:      $k^{i+1} \leftarrow \left( \frac{\check{b} \cdot \hat{k} - \hat{b} \cdot \check{k}}{\hat{k} - \check{k}} \right) \cdot (k^i_{\text{bridge}} - \check{k}) + \hat{b} \cdot \check{k}$
24:    **end if**
25:    $d^{i+1} \leftarrow k^{i+1} \cdot \hat{N}^i + (1 - k^{i+1}) \cdot \hat{d}^i$
26:    $\tilde{x}^{i+1} \leftarrow x + g(d^{i+1}) \cdot \hat{d}^{i+1}$
27:    $i \leftarrow i + 1$
28: **until** Number of queries $= Q$
29: **Return** $\tilde{x}^i$

## 4. Proposed Approach

TtBA determines the perturbation direction $d_k$ at the $i$-th iteration as a weighted combination of the unit current direction $\hat{d}^i$ and its unit normal vector $\hat{N}^i$:

$$d_k = k \cdot \hat{N}^i + (1 - k) \cdot \hat{d}^i, \quad k \in (0, 1], \qquad (5)$$

where the weight parameter $k$ interpolates between the directions $\hat{N}^i$ and $\hat{d}^i$. By identifying a critical bridge threshold $k = k^i_{\text{bridge}}$ (defined below), we can effectively exploit the decision boundary curvature to optimize the perturbation direction efficiently.

Numerous recent studies (Brendel et al., 2018; Cheng et al., 2019; Chen & Gu, 2020; Li et al., 2021; Chen et al., 2020; Reza et al., 2023) have shown that the decision boundary is smooth and locally concave. To further understand the geometric properties of the decision boundary $g(d_k)$, we

iteratively increment $k$ with a step size of 0.01, and estimate $g(d_k)$ using Algorithm 2 in Appendix D. The resulting $g(d_k) - k$ curve when attacking an VGG model using TtBA is plotted in Figure 3. In this figure, $k = k_{\text{local}}$ corresponds to the local minimum of $g(d_k)$, while $k = k^i_{\text{bridge}}$ gives the upper bound of the weight parameter $k$. $k_{\text{local}}$ is hence defined as the weight that determines the local minimum of the decision boundary on the 2D plane spanned by $\hat{d}^i$ and $\hat{N}^i$ at the current iteration of TtBA. Note that when $k > k^i_{\text{bridge}}$, $g(d_k)$ is greater than the decision boundary of the current direction, i.e., $g(d_k) > g(\hat{d}^i)$, resulting in large perturbation as demonstrated in Figure 1. So we first perform a binary search to determine the threshold $k^i_{\text{bridge}}$ and the direction $d^i_{\text{bridge}}$ such that $g(d^i_{\text{bridge}}) = g(\hat{d}^i)$, and then optimize the weight parameter $k$ within the range $(0, k^i_{\text{bridge}}]$ to find $k_{\text{local}}$.

To understand the statistical relationship between $k_{\text{local}}$ and $k^i_{\text{bridge}}$, we analyze the $g(d_k) - k$ curves across 100 images based on both a ViT (Dosovitskiy, 2020) and a VGG (Simonyan & Zisserman, 2015) model, with the distribution of $k_{\text{local}}$ shown in Figure 4. This figure reveals a strong linear relationship between $k_{\text{local}}$ and the threshold $k^i_{\text{bridge}}$ (see theoretical analysis in Appendix B.5). It enables us to directly identify a near-optimal $k_{\text{local}}$ using $k^i_{\text{bridge}}$. For example, as shown in Figure 3, in the 10-th iteration of TtBA, when $k^i_{\text{bridge}} = 0.22$, the local optimum occurs at $k_{\text{local}} = 0.13$, yielding a ratio of $\frac{k_{\text{local}}}{k^i_{\text{bridge}}} = 0.591 \approx 2/3$. By analyzing the distribution of $k_{\text{local}}$ across 100 images from different datasets, as shown in Figure 4, we find that $\frac{k_{\text{local}}}{k^i_{\text{bridge}}}$ is concentrated in the range $[0.5, 0.7]$. Based on this observation, TtBA uses $k^{i+1} = \check{b} \cdot k^i_{\text{bridge}}$ to determine $k_{\text{local}}$ for each iteration, thereby eliminating the need to further search for $k_{\text{local}}$. In this paper, we use $k^{i+1} = 2/3 k^i_{\text{bridge}}$ to efficiently generate the next perturbation (see Appendix G for more details). Driven by this idea, we propose TtBA with its pseudocode shown in Algorithm 1.

In line 3 of Algorithm 1, we first follow the perturbation generation method in HSJA (Chen et al., 2020) and CGBA (Reza et al., 2023) to generate the initial perturbation direction (see Algorithm 4 in Appendix F). In line 5, in each iteration of Algorithm 1, we follow the method in HSJA (Chen et al., 2020) and CGBA (Reza et al., 2023) to generate a normal vector $N^i$ at the current boundary point $\tilde{x}^i$ (see Algorithm 3 in Appendix E). Lines 6–17 perform a binary search to find $k^i_{\text{bridge}}$ in $[0, 1]$ at an accuracy level of $\delta$. Binary search for $k_{\text{bridge}}$ incurs minimal queries for high attack efficiency, as detailed in Appendix C. Empirically, we found that TtBA runs for 57 iterations on average under a 10,000-query budget. Each iteration uses approximately 10 queries for the binary search to achieve $k^i_{\text{bridge}}$ with an accuracy level of $\delta = 0.001$. The total cost of 570 queries ($57 \times 10$) constitutes only 5.7% of the query budget, significantly improving attack performance.

In lines 18–19, $k^i_{\text{bridge}} \geq \hat{k} = 0.2$ indicates that the decision boundary has low curvature and the algorithm is not trapped in any local optimum. In this case, the weight parameter $k^{i+1} = \check{b} \cdot k^i_{\text{bridge}} = 2/3 k^i_{\text{bridge}}$. In lines 20–21, $k^i_{\text{bridge}} \leq \check{k} = 0.1$, indicating that the decision boundary has high curvature and the algorithm is trapped in a local optimum. To escape this local optimum, we increase the weight according to $k^{i+1} = \hat{b} \cdot k^i_{\text{bridge}} = 0.9 k^i_{\text{bridge}}$, forcing $d^{i+1}$ to deviate from the perturbation direction of $\tilde{x}_{\text{local}}$. The effectiveness of this method for escaping local optima is analyzed in Appendix G. As illustrated in Figures 2 and 3, narrow adversarial regions with high decision boundary curvatures tend to have shallow depths (see Appendix B.4 for more details), limiting the optimization of the perturbation magnitude. Increasing $k^{i+1}$ may cause $d^{i+1}$ to deviate from the local best $\tilde{x}_{\text{local}}$, without significantly increasing the $\ell_2$-norm compared to $\tilde{x}_{\text{local}}$.

In lines 23, if $k^i_{\text{bridge}}$ falls within the range $[\check{k} = 0.1, \hat{k} = 0.2]$, the decision boundary is considered to have moderate curvature, potentially causing the optimization process to stuck in a local optimum. To address this, $k^{i+1}$ is chosen as a weighted interpolation between $\hat{b} \cdot k^i_{\text{bridge}}$ and $\check{b} \cdot k^i_{\text{bridge}}$, properly balancing stability and exploration. Specifically, by assigning greater weight to the normal vector, TtBA promotes broader exploration of the perturbation direction, reducing the risk of premature convergence to local optima. In line 25, the new direction $d^{i+1}$ is computed as a weighted combination of the unit current direction $\hat{d}^i$ and the unit normal vector $\hat{N}^i$ using the updated weight $k^{i+1}$. In line 26, a binary search is performed along $d^{i+1}$ (see Algorithm 2 in Appendix D) to identify the boundary point $\tilde{x}^{i+1}$ as the adversarial example of the current iteration.

## 5. Experiments

In this section, we evaluate TtBA's effectiveness through a set of experiments, comparing its performance against state-of-the-art decision-base black-box attacks for both targeted and non-targeted scenarios.

### 5.1. Experiment Settings

**Experiment hardware configuration**. Experiments are conducted using an Intel Xeon Gold 6330 CPU and NVIDIA GeForce RTX 4090 GPU, running PyTorch 2.3.0, Torchvision 0.18.0, and Python 3.11.5.

**Competing approaches**. We compare the performance of TtBA with four state-of-the-art decision-based attacks, including HSJA (Chen et al., 2020), TA (Ma et al., 2021), CGBA, and its variant CGBA-H (Reza et al., 2023), for both non-targeted and targeted scenarios. These methods are commonly used as baselines for decision-based attacks (Ma et al., 2021; Reza et al., 2023). Among them, CGBA and its variant CGBA-H have demonstrated the best attack

performance in non-targeted and targeted settings separately, with the smallest $\ell_2$ perturbation. Therefore, we select these methods for comparison.

**Hyperparameter settings**. We adopt the recommended hyperparameter settings in (Reza et al., 2023) for searching decision boundaries and normal vectors. Specifically, for all four comparative algorithms and TtBA, the decision boundary search tolerance $\tau = 0.0001$. The dimension reduction factor is set to $s = 4$ for the ImageNet dataset, and $s = 1$ for all other datasets. In TtBA, search tolerance $\delta = 0.001$ for determining $k^i_{\text{bridge}}$. According to the parameter sensitive analysis in Appendix G, the thresholds $\check{k} = 0.1$ and $\hat{k} = 0.2$, and the bridge coefficients $\hat{b} = 0.9$ and $\check{b} = 2/3$.

**Benchmark datasets and models**. To assess the effectiveness and scalability of TtBA, we chose five datasets and 9 typical models listed below:

1. MNIST-CNN: The MNIST dataset (LeCun et al., 1998), and a benchmark 7-layer MNIST-CNN (Cheng et al., 2020; Chen & Gu, 2020) that achieved 99.4% accuracy after training;

2. FASHION-MNIST-CNN: The FASHION MNIST dataset (Xiao et al., 2017), and a benchmark 7-layer FASHION-MNIST-CNN (Cheng et al., 2020; Chen & Gu, 2020) that achieved 91.0% accuracy after training.

3. CIFAR10-CNN: The CIFAR10 dataset (Krizhevsky et al., 2009), and a benchmark 7-layer CIFAR10-CNN (Cheng et al., 2020; Chen & Gu, 2020) that achieved 82.5% accuracy after training;

4. CIFAR100-ViT: The CIFAR100 dataset (Krizhevsky et al., 2009), and a well-known ViT model (Dosovitskiy, 2020) that achieved 89.9% accuracy after training;

5. The ImageNet dataset (Deng et al., 2009), and five prominent machine learning models: ImageNet-VGG19 (Simonyan & Zisserman, 2015), ImageNet-ResNet50 (He et al., 2016), ImageNet-ViT (Dosovitskiy, 2020), ImageNet-EfficientNet (Tan & Le, 2019), and ImageNet-Inception (Szegedy et al., 2016).

### 5.2. Main Results

**Performance metrics**. Following state-of-the-art black-box attacks (Brendel et al., 2018; Li et al., 2021; Reza et al., 2023), we adopt the $\ell_2$ norm to measure the perturbation strength. For each model, we randomly select 1000 images from the test dataset. Table 1 presents the average and median $\ell_2$ norms achieved by all competing approaches under query budgets of 2,000, 5,000, and 10,000, for both non-targeted and targeted attacks.

| Datasets | Attacks | Non-targeted attacks | | | Targeted attacks | | |
|---|---|---|---|---|---|---|---|
| | | 2,000 QUE | 5,000 QUE | 10,000 QUE | 2,000 QUE | 5,000 QUE | 10,000 QUE |
| MNIST | HSJA | 2.921(1.785) | 1.609(1.443) | 1.366(1.307) | 2.956(2.712) | 2.275(2.079) | 1.980(1.869) |
| CNN | TA | 2.983(1.799) | 1.588(1.417) | 1.345(1.281) | 2.819(2.634) | 2.169(2.001) | 1.901(1.831) |
| | CGBA | 2.689(1.825) | 1.978(1.538) | 1.436(1.410) | 3.247(2.904) | 2.465(2.223) | 2.095(2.021) |
| | CGBA-H | 2.741(1.864) | 1.631(1.445) | 1.373(1.317) | 2.977(2.846) | 2.343(2.102) | 2.059(1.862) |
| | TtBA | **2.667(1.754)** | **1.505(1.362)** | **1.288(1.213)** | **2.841(2.659)** | **2.156(2.017)** | **1.884(1.786)** |
| FASHION | HSJA | 1.127(0.954) | 0.778(0.748) | 0.683(0.665) | 1.741(1.689) | 1.256(1.294) | 1.097(1.130) |
| -MNIST | TA | 1.171(0.998) | 0.779(0.730) | 0.665(0.646) | 1.729(1.691) | 1.239(1.222) | 1.017(1.100) |
| CNN | CGBA | 1.059(0.908) | 0.853(0.704) | 0.781(0.640) | 1.729(1.704) | 1.263(1.291) | 1.110(1.156) |
| | CGBA-H | 1.021(0.984) | 0.783(0.756) | 0.690(0.665) | 1.681(1.636) | 1.261(1.259) | 1.101(1.115) |
| | TtBA | **0.896(0.873)** | **0.674(0.672)** | **0.603(0.601)** | **1.642(1.605)** | **1.196(1.208)** | **1.050(1.084)** |
| CIFAR10 | HSJA | 0.541(0.450) | 0.300(0.251) | 0.226(0.196) | 1.634(1.081) | 0.622(0.468) | 0.410(0.332) |
| CNN | TA | 0.552(0.449) | 0.289(0.245) | 0.208(0.190) | 1.514(0.823) | 0.554(0.401) | 0.348(0.301) |
| | CGBA | **0.350(0.279)** | 0.233(0.189) | 0.181(0.152) | 1.121(0.794) | 0.440(0.393) | 0.317(0.298) |
| | CGBA-H | 0.383(0.309) | 0.235(0.190) | 0.188(0.153) | 0.872(0.602) | 0.452(0.353) | 0.338(0.284) |
| | TtBA | 0.371(0.294) | **0.227(0.188)** | **0.180(0.151)** | **0.817(0.595)** | **0.415(0.343)** | **0.308(0.263)** |
| CIFAR | HSJA | 6.322(4.691) | 2.627(2.015) | 1.750(1.257) | 42.66(40.43) | 19.78(17.18) | 10.88(8.074) |
| -100 | TA | 5.825(4.376) | 2.327(1.925) | 1.350(1.187) | 40.38(38.42) | 18.04(16.53) | 9.761(7.217) |
| ViT | CGBA | **2.086(1.430)** | 1.097(0.847) | 0.744(0.613) | 45.15(44.12) | 11.96(7.399) | 4.480(2.397) |
| | CGBA-H | 2.165(1.500) | 1.130(0.862) | 0.792(0.678) | 20.66(18.41) | 8.228(4.571) | **4.327**(2.172) |
| | TtBA | 2.114(1.521) | **1.079(0.845)** | **0.734(0.603)** | **20.15(15.18)** | **8.154(4.330)** | 4.464(**2.160**) |
| ImageNet | HSJA | 27.33(16.95) | 13.39(7.018) | 7.840(4.023) | 72.01(61.64) | 55.80(48.15) | 46.25(41.91) |
| ResNet50 | TA | 23.38(14.99) | 12.04(6.302) | 7.251(3.539) | 68.95(59.52) | 52.91(46.12) | 44.34(39.93) |
| | CGBA | 18.58(12.79) | 10.10(5.099) | 5.838(2.499) | 76.38(70.77) | 69.25(64.89) | 58.33(52.44) |
| | CGBA-H | 17.79(9.628) | 10.63(4.530) | 6.203(2.542) | 59.25(55.30) | 45.99(42.75) | 34.35(30.35) |
| | TtBA | **12.33(7.462)** | **6.32(2.804)** | **3.448(1.700)** | **57.21(53.19)** | **42.38(37.83)** | **27.83(24.54)** |
| ImageNet | HSJA | 8.117(5.409) | 4.397(2.513) | 3.256(1.997) | 71.28(67.48) | 50.92(36.57) | 32.77(19.81) |
| VGG19 | TA | 6.592(4.433) | 3.737(2.028) | 2.651(1.539) | 63.59(61.12) | 40.18(32.17) | 26.91(17.22) |
| | CGBA | 3.810(2.184) | 1.912(1.119) | 1.174(0.750) | 80.00(77.99) | 62.89(60.84) | 40.72(33.11) |
| | CGBA-H | 3.783(2.162) | 1.841(1.179) | 1.253(0.740) | 54.09(46.12) | 29.05(21.44) | 14.32(10.59) |
| | TtBA | **3.650(2.150)** | **1.834(1.105)** | **1.166(0.737)** | **46.89(40.99)** | **25.85(20.14)** | **12.26(8.420)** |
| ImageNet | HSJA | 12.24(9.242) | 6.513(4.233) | 4.071(2.761) | 33.73(29.98) | 15.91(15.01) | 7.899(7.848) |
| ViT | TA | 8.926(6.115) | 4.792(3.101) | 2.872(2.002) | 28.13(24.34) | 13.38(10.12) | 5.647(4.649) |
| | CGBA | 4.515(3.086) | 2.338(1.571) | 1.588(1.058) | 35.72(30.62) | 13.96(10.19) | 5.829(4.757) |
| | CGBA-H | 4.554(3.078) | 2.311(1.651) | 1.582(1.131) | 23.23(20.75) | 9.614(7.794) | 4.862(3.944) |
| | TtBA | **4.508(3.052)** | **2.294(1.558)** | **1.512(1.044)** | **22.20(19.64)** | **9.205(7.616)** | **4.841(3.817)** |
| ImageNet | HSJA | 16.25(11.18) | 9.288(4.980) | 5.816(3.021) | 77.46(74.88) | 66.38(63.01) | 51.29(47.72) |
| Efficient | TA | 13.11(8.728) | 6.986(4.034) | 4.112(2.429) | 73.00(68.15) | 62.77(58.26) | 49.28(44.36) |
| | CGBA | 9.956(5.719) | 5.696(2.522) | 3.161(1.456) | 79.44(75.73) | 68.52(61.24) | 52.82(44.16) |
| | CGBA-H | 9.542(6.373) | 4.836(2.958) | 2.688(1.979) | 61.98(56.37) | 42.34(38.71) | 28.83(25.39) |
| | TtBA | **6.451(4.400)** | **3.359(2.136)** | **2.008(1.224)** | **54.85(50.73)** | **36.15(30.59)** | **21.68(15.81)** |
| ImageNet | HSJA | 14.50(9.801) | 7.823(4.565) | 4.935(2.820) | 75.51(72.86) | 64.54(61.28) | 50.37(46.89) |
| Inception | TA | 13.23(8.012) | 6.907(3.854) | 3.839(2.233) | 71.26(67.54) | 60.83(57.36) | 48.21(43.52) |
| | CGBA | 11.54(4.402) | 5.950(1.935) | 3.185(1.072) | 80.47(77.25) | 67.99(65.07) | 53.69(48.67) |
| | CGBA-H | 10.88(4.882) | 5.095(1.960) | 2.474(1.092) | 56.46(51.08) | 42.22(36.61) | 27.96(22.60) |
| | TtBA | **8.599(4.297)** | **3.771(1.765)** | **1.961(0.953)** | **55.16(53.13)** | **38.24(34.55)** | **24.19(18.18)** |

*Table 1.* Average (median) $\ell_2$ norm of perturbation for targeted and non-targeted black-box attacks under different query budgets (QUE).

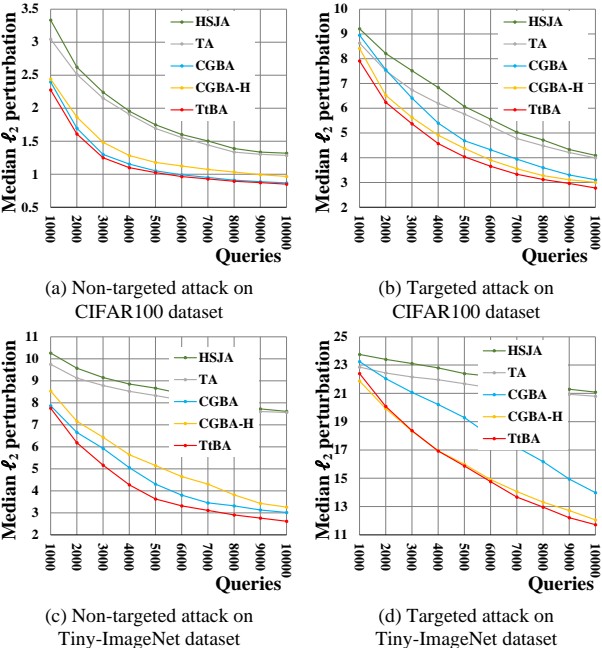

(a) Non-targeted attack on
CIFAR100 dataset

(b) Targeted attack on
CIFAR100 dataset

(c) Non-targeted attack on
Tiny-ImageNet dataset

(d) Targeted attack on
Tiny-ImageNet dataset

*Figure 5.* Median $\ell_2$ perturbation against adversarially-trained WideResNet models.

**Results**. For non-targeted attacks in Table 1, our proposed TtBA achieves the best performance on 7 out of 9 models, ranging from 2000 to 10,000 queries. On the ImageNet dataset, TtBA reduces the $\ell_2$ perturbation by over 20% compared to other methods for the ResNet50 and Efficient models. On the CIFAR10-CNN and CIFAR100-ViT models, CGBA outperforms TtBA in terms of both average and median $\ell_2$ norm at 2,000 queries. However, TtBA surpasses CGBA at 5,000 and 10,000 queries for both models. In targeted attacks, TtBA surpasses other methods in nearly all tested scenarios, covering 9 models and various query budgets. The only exception occurs with the CIFAR100-ViT model at 10,000 queries, where CGBA-H achieves a marginally lower average $\ell_2$ norm (4.327 versus 4.464). This result shows that TtBA outperforms other methods, especially for targeted attack settings.

The perturbed images generated by TtBA for different query budgets (QUE) are shown in Figure 6. In this figure, perturbation directions are normalized to the $[0, 1]$ range to illustrate how they diminish as the query budget increases, starting from random noise in non-targeted attacks and a target image in targeted one.

### 5.3. Results on Adversarially Trained Models

Defense methods based on adversarial training play a crucial role in enhancing model robustness by significantly lowering the success rates of hard-label attacks (Chen & Gu, 2020; Chakraborty et al., 2021). We assess TtBA's ef-

fectiveness on state-of-the-art adversarially trained models developed in (Wang et al., 2023). These models serve as a highly relevant benchmark due to their focus on adversarial robustness, making them ideal for assessing the performance of TtBA. In our experiments, we test specifically the WideResNet models (Wang et al., 2023), which were trained using techniques in (Zagoruyko, 2016) to achieve strong robustness. We compare TtBA against other leading attack methods, including HSJA, TA, CGBA, and CGBA-H, on the CIFAR-100 and Tiny-ImageNet datasets. For each dataset, we randomly select 500 images to conduct targeted and non-targeted attack experiments, adhering to a query budget limit of 10,000 queries. Specifically, we adopt the WideResNet WRN-70-16 model from (Wang et al., 2023) for CIFAR-100 and the WRN-28-10 model for Tiny-ImageNet. Past studies show that these models have strong defensive performance on the respective datasets (Wang et al., 2023).

To assess the general applicability of TtBA, we follow identical hyper-parameter settings introduced in Section 5.1. The median $\ell_2$ curve with respect to different numbers of queries is presented in Figure 5. The results in Figure 5-(a) and (b) demonstrate that, on the CIFAR100 dataset, TtBA outperforms all competing methods for both targeted and non-targeted attacks. For the tiny-ImageNet dataset, Figure 5-(c) shows that TtBA achieves the best performance in non-targeted attacks. However, for targeted attacks, as depicted in Figure 5-(d), CGBA-H performs the best within the query budget range of 1,000 to 4,000. Beyond 4,000 queries, TtBA surpasses CGBA-H.

## 6. Conclusion and Future Work

In this paper, we proposed *Two-third Bridge Attack* (TtBA), a novel and efficient decision-based black-box adversarial attack method. Our method introduces the concept of bridge direction, which effectively combines the current perturbation direction with its normal vector, guided by the weight parameter $k_{\text{bridge}}$. We identified and successfully addressed the challenge of local optima in regions with high decision boundary curvatures, significantly enhancing the attack's effectiveness. Further, through theoretical analysis and extensive experimentation, we demonstrated that using $2/3k_{\text{bridge}}$ consistently yields near-optimal results in minimizing the $\ell_2$ distance of adversarial examples. Extensive evaluation across multiple datasets (MNIST, FASHION-MNIST, CIFAR10, CIFAR100, and ImageNet) and 9 deep learning models demonstrated that TtBA can consistently outperform state-of-the-art methods in both targeted and non-targeted attack scenarios. Our work not only advances the field of adversarial machine learning but also provides valuable insights into the geometric properties of decision boundaries through the novel $k_{\text{bridge}}$ metric.

Future research directions for TtBA are promising in sev-

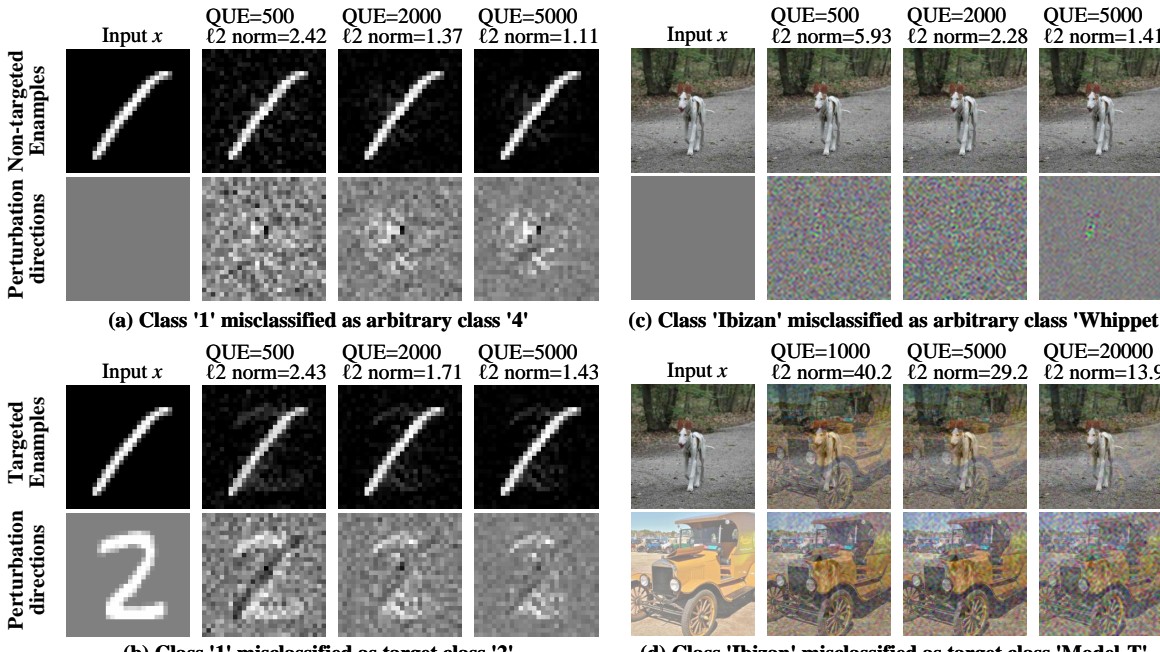

Figure 6. Adversarial examples and corresponding perturbation directions generated by TtBA for different query budgets.

eral avenues. First, adapting TtBA to score-based attack scenarios could leverage probability outputs to significantly reduce query complexity. Second, incorporating TtBA into multi-objective optimization frameworks could enhance real-world deployability by jointly optimizing perturbation imperceptibility, cross-model transferability, and query efficiency. The geometric insights developed from the decision boundary analysis in this paper could extend beyond traditional attack scenarios, potentially benefiting adversarial machine learning tasks in areas such as speech recognition, text classification, and object detection.

## Impact Statement

This paper presents work that aims to advance the field of AI security, specifically in understanding and improving black-box adversarial attacks against image classification models. While our research could potentially be used to enhance the robustness of AI systems through better understanding of their vulnerabilities, we acknowledge that adversarial attack techniques might have dual-use implications. Our proposed method can potentially be extended beyond image classification to other domains including speech recognition, text classification, and object detection models. We believe this research contributes to the broader goal of developing more reliable and secure AI systems across various AI applications, which is crucial as these technologies become increasingly integrated into society.

## Acknowledgments

This work was partially supported by BUPT Excellent Ph.D. Students Foundation, foundation number is CX20241003. The authors express their gratitude to the anonymous reviewers for their insightful and constructive feedback on the initial version of this paper.

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

# A. Appendix Introduction

In real-world applications, lower perturbation strength makes adversarial examples less noticeable to human observers. Therefore, decision-based attacks aim to deceive the target DNN (i.e., image classifier) with the minimal perturbation strength, subject to a pre-determined query budget (Brendel et al., 2018). To enhance attack efficiency, decision-based attacks have focused on exploring *perturbation directions* and the corresponding *decision boundary* to optimize adversarial examples (Brendel et al., 2018; Chen & Gu, 2020; Reza et al., 2023). A perturbation direction is a pixel-wise vector guiding adversarial modifications. Its decision boundary defines the minimal perturbation magnitude required to mislead the model along the perturbation direction. Thus the direction with low decision boundary is highly desirable. As illustrated in Figure 1, $x$ is the original image, and the boundary of the orange region represents the decision boundary of a DNN in image classification. Points $\tilde{x}^i$ and $\tilde{x}^i_{\text{bridge}}$ in Figure 1 serve as boundary points along the perturbation directions $\hat{d}^i$ and $d^i_{\text{bridge}}$, respectively.

The *curvatures* of decision boundaries have strong impact on perturbation optimization (Ma et al., 2021; Reza et al., 2023). In particular, targeted attacks often create narrower adversarial regions with higher decision boundary curvatures than non-targeted attacks, making perturbation optimization more challenging than non-targeted attacks (Reza et al., 2023). As shown in Figure 2-(c), the adversarial region contracts in targeted attacks, making it difficult to find effective adversarial examples, seriously hurting attack effectiveness and query-efficiency (Reza et al., 2023). Thus, efficiently optimizing perturbations within narrow adversarial regions is a crucial practical challenge (Reza et al., 2023). Overcoming this obstacle is key to advancing the effectiveness of decision-based attacks.

# B. Geometric Analysis of Decision Boundary

### B.1. Define the HyperSurface of Decision Boundary

The decision boundary of the target DNN model $f$ can be characterized as a hyperSurface $S$ associated with an input $x$. Specifically, $S$ is the set of all points $\tilde{x}$ that lie exactly on the boundary between the class of $x$ and other classes. For any direction $d \in \mathbb{R}^{C \times W \times H}$, there exists a unique boundary point $\tilde{x}$ along $d$, determined by the minimal perturbation required to cross the decision boundary. Formally, $S$ can be defined as:

$$S = \left\{ \tilde{x} = x + g(d) \cdot \frac{d}{\|d\|_2} \,\middle|\, \forall d \neq 0,\ I(\tilde{x}) = 1 \right\}, \tag{6}$$

where $I(\cdot)$ is the indicator function that determines whether $\tilde{x}$ lies in the adversarial region, $g(d)$ is the decision boundary along direction $d$, $\frac{d}{\|d\|_2}$ is the unit direction vector. HSJA (Chen et al., 2020) proves that there exists only one boundary point on $S$ along any direction $d$. Furthermore, since the hyperSurface $S$ is smooth with respect to $d$, the decision boundary $g(d)$ is also smooth with respect to $d$.

### B.2. Local Taylor Expansion of Decision Boundary

Since studying the properties of $S$ in the high-dimensional input space $C \times W \times H$ is challenging, existing research such as (Chen et al., 2020; Ma et al., 2021; Reza et al., 2023) focused on studying the curve at the intersection between the decision boundary and the 2D plane defined by the current perturbation vector $\hat{d}^i$ and the normal vector $\hat{N}^i$. This 2D reduction is effective because $\hat{N}^i$ indicates the direction of locally optimal perturbation, making the search for new directions more efficient. For simplicity, we treat the decision boundary $G(k) = g(k \cdot \hat{N}^i + (1-k) \cdot \hat{d}^i)$ as a function of $k$ and analyze the properties of $G(k)$ on this 2D plane. We express $G(k)$ as a Taylor expansion around the initial point $k = 0$:

$$G(k) = g(\hat{d}^i) + ak + \frac{1}{2}bk^2 + \frac{1}{6}ck^3 + \mathcal{O}(k^4), \tag{7}$$

- $a = \frac{\partial G}{\partial k}\big|_{k=0} < 0$ is the first-order derivative, representing the initial slope of the boundary along $\hat{N}^i$,

- $b = \frac{\partial^2 G}{\partial k^2}\big|_{k=0} > 0$ is the second-order derivative, dominating local curvature,

- $c = \frac{\partial^3 G}{\partial k^3}\big|_{k=0}$ is the third-order derivative, describing the curvature variation rate,

- $\mathcal{O}(k^4)$ denotes higher-order terms.

To find $k_{\text{bridge}}^i$, we set

$$G(k) = g(\hat{d}^i) + ak + \frac{1}{2}bk^2 + \frac{1}{6}ck^3 + \mathcal{O}(k^4) = g(\hat{d}^i).$$

By omitting the third-order term $\frac{1}{6}ck^3$ and higher-order terms, the equation simplifies to

$$G(k) = g(\hat{d}^i) + ak + \frac{1}{2}bk^2 = g(\hat{d}^i).$$

Moving $g(\hat{d}^i)$ to the right-hand side yields

$$ak + \frac{1}{2}bk^2 = 0.$$

This equation has two solutions:

$$k = 0 \quad \text{and} \quad a + \frac{1}{2}bk = 0,$$

which gives

$$k = -\frac{2a}{b}.$$

Since $k_{\text{bridge}}^i > 0$, we obtain

$$k_{\text{bridge}}^i = -\frac{2a}{b}. \tag{8}$$

### B.3. Relationship between Curvature and $k_{\text{bridge}}$

At $k = 0$, the curvature $\kappa$ of the decision boundary in the 2D plane is derived from the Taylor expansion coefficients as:

$$\kappa = \frac{|f''(k)|}{(1 + (f'(k))^2)^{3/2}} = \frac{b}{(1 + a^2)^{3/2}}. \tag{9}$$

This revision simplifies the wording slightly, maintains the mathematical meaning, and improves the flow of the sentence. When higher-order terms are negligible (i.e., $|a| \ll 1$ and $|ck^3| \ll |bk^2|$), the curvature can be approximated as:

$$\kappa \approx b \approx -\frac{2a}{k_{\text{bridge}}^i}, \quad \text{with} \quad \kappa \propto \frac{1}{k_{\text{bridge}}^i}. \tag{10}$$

This approximation is valid for small attack steps within the neighborhood of $k$. It implies that as the curvature of the decision boundary $G(k)$ increases, $k_{\text{bridge}}^i$ decreases. Thus, $k_{\text{bridge}}^i$ serves as an indicator of the decision boundary's curvature.

### B.4. Minimum Points and Values of Decision Boundary

CRITICAL POINTS AND MINIMUM DECISION BOUNDARY

The first-order derivative of $G(k)$ is:

$$G'(k) = a + bk + \frac{1}{2}ck^2 + \mathcal{O}(k^3). \tag{11}$$

Setting $G'(k) = 0$ to find critical points, we solve:

$$a + bk + \frac{1}{2}ck^2 \approx 0 \quad \text{(neglecting } \mathcal{O}(k^3)\text{)}. \tag{12}$$

Quadratic Approximation Case ($c = 0$): When higher-order terms are negligible ($c \approx 0$), the critical point simplifies to:

$$k_{\min} = -\frac{a}{b}. \tag{13}$$

The second derivative $G''(k) = b + ck + \mathcal{O}(k^2)$ must satisfy $G''(k_{\min}) > 0$ for a local minimum. Under $c \approx 0$, this reduces to $b > 0$, consistent with positive curvature.

From the curvature formula $\kappa = \frac{b}{(1+a^2)^{3/2}}$ (Eq. (9)), two regimes emerge:

1. **Small Slope ($|a| \ll 1$):** Here, $\kappa \approx b$, and Eq. (13) becomes:

$$k_{\min} \approx -\frac{a}{\kappa}. \tag{14}$$

For fixed $a$, increasing $\kappa$ (i.e., sharper curvature) drives $k_{\min} \to 0$. The minimum value is:

$$G(k_{\min}) \approx g(\hat{d}^i) - \frac{a^2}{2\kappa}. \tag{15}$$

Larger $\kappa$ reduces the subtraction term, resulting in a *larger $G(k_{\min})$*.

2. **Non-Negligible Slope ($|a| \sim 1$):** The full curvature formula $\kappa = \frac{b}{(1+a^2)^{3/2}}$ implies $b = \kappa(1 + a^2)^{3/2}$. Substituting into $k_{\min}$:

$$k_{\min} = -\frac{a}{\kappa(1 + a^2)^{3/2}}. \tag{16}$$

Even with moderate $a$, increasing $\kappa$ still forces $k_{\min} \to 0$, and $G(k_{\min})$ grows as $G(k_{\min}) \approx g(\hat{d}^i) - \frac{a^2}{2\kappa(1+a^2)^{3/2}}$.

- **High Curvature ($\kappa \uparrow$):** The decision boundary is sharply curved near input $x$. To reach the boundary, adversarial perturbations require smaller steps ($k_{\min} \downarrow$) but larger magnitudes ($G(k_{\min}) \uparrow$), as the boundary "retreats" rapidly from the original class.

- **Low Curvature ($\kappa \downarrow$):** The boundary is flat, allowing larger steps ($k_{\min} \uparrow$) with smaller perturbations ($G(k_{\min}) \downarrow$).

Including the cubic term ($c \neq 0$), Eq. (12) becomes quadratic:

$$\frac{1}{2}ck^2 + bk + a = 0. \tag{17}$$

The solution is:

$$k_{\min} = \frac{-b \pm \sqrt{b^2 - 2ac}}{c}. \tag{18}$$

For small $c$, a Taylor expansion in $c/b^2$ gives:

$$k_{\min} \approx -\frac{a}{b} + \frac{a^2 c}{2b^3} + \mathcal{O}(c^2). \tag{19}$$

The cubic term introduces an offset proportional to $c$, but the dominant term $-a/b$ still ensures $k_{\min} \propto 1/\kappa$ when $\kappa \gg 1$.

### B.5. Linear Relationship Between $k_{\min}$ and $k_{\text{bridge}}$

Building on the analysis of the decision boundary's curvature $\kappa$ and the minimum point $k_{\min}$, we now investigate the relationship between $k_{\min}$ and the parameter $k_{\text{bridge}}^i$. We demonstrate that under the quadratic approximation, $k_{\min}$ and $k_{\text{bridge}}^i$ exhibit an approximately linear relationship, governed by the curvature $\kappa$.

1. Minimum Point $k_{\min}$: From Eq. (13), under quadratic approximation ($c = 0$):

$$k_{\min} \approx -\frac{a}{b},$$

where $a = \partial G/\partial k|_{k=0}$ and $b = \partial^2 G/\partial k^2|_{k=0} \approx \kappa$ when $|a| \ll 1$.

2. Bridge weight parameter $k_{\text{bridge}}$ from Eq. (8):

$$k_{\text{bridge}}^i \approx -\frac{2a}{b}.$$

LINEAR RELATIONSHIP DERIVATION

From Eq. (13) and Eq. (8) , we obtain:

$$k_{\min} \approx \frac{1}{2} \cdot k_{\text{bridge}}^i.$$

This reveals a *linear proportionality* between $k_{\min}$ and $k_{\text{bridge}}^i$:

$$k_{\min} \propto k_{\text{bridge}}^i. \tag{20}$$

## C. Complexity of Binary Search for $k_{\text{bridge}}$

In lines 6-17 of Algorithm 1, we perform a binary search to identify the bridge parameter $k_{\text{bridge}}^i$ that aligns the intermediate direction $d_{\text{mid}}$ (Line 9) with the current direction $d^i$'s decision boundary. While this process requires query budget, its computational cost is minimal compared to its strategic benefits. Under the standard 10,000-query setting (Chen & Gu, 2020), TtBA completes 57 iterations on average, with query allocation distributed across four components:

1. Initial perturbation (Algorithm 4): about 10 queries for baseline direction generation.

2. Normal vector estimation (Line 5): $\lceil 30\sqrt{i} \rceil$ queries per iteration for boundary geometry analysis, where $i$ represent the $i$-th optimization iteration in Algorithm 1.

3. Bridge parameter binary search (Lines 6-17): approximately 10 queries/iteration at precision $\delta = 0.001$ ($\approx 2^{-10}$)

4. Decision boundary binary search (Line 26): approximately 14 queries/iteration for $\tau = 0.0001$ ($\approx 2^{-14}$) tolerance.

The resulting queries for each iteration is:

$$q_i = 24 + \lceil 30\sqrt{i} \rceil. \tag{21}$$

Consequently, the total number queries across $n$ iterations of binary search is bounded by:

$$Q_n = 10 + \sum_{i=1}^{n} q_i \leq 10 + \int_1^n (24 + 30\sqrt{x})\, dx = 10 + \left[24x + 20x^{3/2}\right]_1^n = 24n + 20n^{3/2} - 34. \tag{22}$$

Solving $24n + 20n^{3/2} - 34 = 10^4$ yields $n \approx 57$ iterations. The total cost of 570 queries ($57 \times 10$) for $k_{\text{bridge}}$ binary search constitutes only 5.7% of the query budget.

## D. Binary Search of Decision Boundary

In Algorithm 2, the binary search process begins by initializing the low boundary point $\tilde{x}_{\text{low}} \leftarrow x$ in the non-adversarial region and the high boundary point $\tilde{x}_{\text{high}} \leftarrow x + d$ in the adversarial region. In lines 4-11, the algorithm iteratively checks whether the midpoint $\tilde{x}_{\text{mid}} \leftarrow (\tilde{x}_{\text{low}} + \tilde{x}_{\text{high}})/2$ lies in the adversarial region and updates $\tilde{x}_{\text{low}}$ or $\tilde{x}_{\text{high}}$ accordingly until the search tolerance is met. Finally, in line 12, it returns the decision boundary value $\|\tilde{x}_{\text{high}} - x\|_2$.

## E. Normal Vector Generation

We follow the CGBA (Reza et al., 2023) normal vector estimation method (Algorithm 3) for decision boundary characterization. Given a boundary point $\tilde{x}$ in line 1, we initialize a normal vector $N \leftarrow \mathbf{0}^{C \times W \times H}$ in line 3. For each query $q \in [1, Q_N]$: 1) Generate low-dimensional Gaussian noise $d_{\text{temp}} \sim \mathcal{N}(0, I)$ in reduced space $\mathbb{R}^{C \times \frac{W}{s} \times \frac{H}{s}}$ (Line 5); 2) Upsample to image space via 2D inverse discrete cosine transform $d_{\text{Gaussian}} \leftarrow \text{IDCT}_2(d_{\text{temp}})$ (Line 6), where $\text{IDCT}_2(\cdot)$ refers to the 2D Inverse Discrete Cosine Transform, which reconstructs spatial-domain data from its frequency-domain representation; 3) Test adversarial direction by perturbing $\tilde{x}$ with $r_N \cdot \hat{d}_{\text{Gaussian}}$ (Line 7); 4) Accumulate direction using $N \leftarrow N \pm \hat{d}_{\text{Gaussian}}$ based on the indicator $I(\tilde{x} + r_N \hat{d}_{\text{Gaussian}})$ (Lines 8-10). The final normalized normal vector $\hat{N}$ is returned after $Q_N$ queries (Line 12).

---

**Algorithm 2** Decision Boundary Binary Search

---

1: **Input:** Original image $x$, direction $d$, indicator $I(\cdot)$;
2: **Output:** Decision boundary $g(d)$ for direction $d$;
3: **Initialization:** Set low boundary point $\tilde{x}_{\text{low}} \leftarrow x$, high boundary point $\tilde{x}_{\text{high}} \leftarrow x + d$, search tolerance $\tau \leftarrow 0.0001$;
4: **repeat**
5:     $\tilde{x}_{\text{mid}} \leftarrow \frac{\tilde{x}_{\text{low}} + \tilde{x}_{\text{high}}}{2}$
6:     **if** $I(\tilde{x}_{\text{mid}}) = 1$ **then**
7:         $\tilde{x}_{\text{high}} \leftarrow \tilde{x}_{\text{mid}}$
8:     **else**
9:         $\tilde{x}_{\text{low}} \leftarrow \tilde{x}_{\text{mid}}$
10:    **end if**
11: **until** $\|\tilde{x}_{\text{high}} - \tilde{x}_{\text{low}}\|_2 \leq \tau$
12: **Return** $\|\tilde{x}_{\text{high}} - x\|_2$ as $g(d)$.

---

**Algorithm 3** Get Normal Vector

---

1: **Input:** Original image $x$, indicator function $I(\cdot)$, a boundary point $\tilde{x}$, query budget for normal vector $Q_N$, dimension reduce factor $s$, and normal vector radius $r_N = 0.0003$;
2: **Output:** A normal vector $N$ at current boundary point $\tilde{x}$;
3: **Initialization:** $N \leftarrow (0, ..., 0)^{\text{C} \times \text{W} \times \text{H}}$,
4: **for** query $q = 1$ **to** $Q_N$ **do**
5:     $d_{\text{temp}} \leftarrow (z_{1,1,1}, ..., z_{\text{C}, \frac{W}{s}, \frac{H}{s}}), z_{\text{c,w,h}} \sim \mathcal{N}(0,1)$
6:     $d_{\text{Gaussian}} \leftarrow \text{IDCT}_2(d_{\text{temp}})$
7:     **if** $I(\tilde{x} + r_N \cdot \hat{d}_{\text{Gaussian}}) = 1$ **then**
8:         $N \leftarrow N + \hat{d}_{\text{Gaussian}}$
9:     **else**
10:        $N \leftarrow N - \hat{d}_{\text{Gaussian}}$
11:    **end if**
12: **end for**
13: **Return** $N$

---

## F. Initial Perturbation Generation

We adopt the CGBA framework (Reza et al., 2023) to compute the initial perturbation direction (Algorithm 4). For targeted attacks (Lines 3-4), the direction is derived as $d_{\text{init}} = x_{\text{target}} - x$. In non-targeted scenarios (Lines 6-19), Gaussian noise $d_{\text{Gaussian}} \sim \mathcal{N}(0, I)$ is iteratively sampled (Line 7), scaled by $0.02q$ (Line 8, where $q$ is the query index and $\hat{\cdot}$ denotes normalization), with early termination if $I(x + d_{\text{init}}) = 1$ (Lines 9-11). If $Q/10$ queries fail (Line 13), perturbations $\mathbf{1}^{C \times W \times H} - x$ (Line 14) or $\mathbf{0}^{C \times W \times H} - x$ (Line 16) are attempted. The direction is then calibrated via $d_{\text{init}} \leftarrow g(d_{\text{init}}) \cdot \hat{d}_{\text{init}}$ (Line 20), where $g(\cdot)$ implements boundary projection through binary search (Reza et al., 2023).

## G. Parameter Sensitivity Analysis

Our TtBA algorithm summarized in Algorithm 1 introduces several parameters, including the local optima thresholds $\check{k} = 0.1$ and $\hat{k} = 0.2$, as well as the bridge coefficients $\hat{b} = 0.9$ and $\check{b} = 2/3$. The corresponding parameter settings are determined based on preliminary experiments conducted on the ImageNet-VGG19 model. In this appendix, we show that the performance of TtBA is not sensitive to these parameter settings.

To evaluate the impact of different parameter settings, we conduct a sensitivity analysis. Specifically, we vary the bridge coefficients $\hat{b}$ and $\check{b}$ (Table 2) and the local optima thresholds $\check{k}$ and $\hat{k}$ (Table 3) across a range of values. For each setting, performance is assessed using the area under the median $\ell_2$-query curve (AUC). AUC is calculated using the formula: $\text{AUC} = \sum_{i=1}^{n} \ell_2(Q_i)$ where $\ell_2(Q_i)$ represents the median $\ell_2$-norm obtained after $Q_i$ queries, and $n = 10,000$ is the total query budget. In adversarial attacks, a lower AUC demonstrates a smaller perturbation and better attack effect.

We conduct experiments on the MNIST-CNN, CIFAR10-CNN, CIFAR100-ViT, and ImageNet-VGG models for both

---

**Algorithm 4** Get Initial Perturbation Direction

---

1: **Input:** Original image $x$, indicator $I(\cdot)$, query budget $Q$, target image $x_{\text{target}}$ for targeted attack;
2: **Output:** An initial perturbation direction $d_{\text{init}}$;
3: **if** Targeted attack **then**
4:     $d_{\text{init}} \leftarrow x_{\text{target}} - x$
5: **else if** Non-targeted attack **then**
6:     **for** query $q = 1$ **to** $\frac{Q}{10}$ **do**
7:         $d_{\text{Gaussian}} \leftarrow (z_{1,1,1}, ..., z_{\text{C,W,H}}), z_{c,w,h} \sim \mathcal{N}(0,1)$
8:         $d_{\text{init}} \leftarrow 0.02 \cdot q \cdot \hat{d}_{\text{Gaussian}}$
9:         **if** $I(x + d_{\text{init}}) = 1$ **then**
10:            break
11:         **end if**
12:     **end for**
13:     **if** $q \geq \frac{Q}{10}$ **then**
14:         $d_{\text{init}} \leftarrow (1, ..., 1)^{\text{C}\times\text{W}\times\text{H}} - x$
15:         **if** $I(x + d_{\text{init}}) = -1$ **then**
16:            $d_{\text{init}} \leftarrow (0, ..., 0)^{\text{C}\times\text{W}\times\text{H}} - x$
17:         **end if**
18:     **end if**
19: **end if**
20: $d_{\text{init}} \leftarrow g(d_{\text{init}}) \cdot \hat{d}_{\text{init}}$
21: **Return** $d_{\text{init}}$

---

| Dataset | Parameter $\hat{b}$ | $\hat{b} = 0.9$ | $\hat{b} = 0.9$ | $\hat{b} = 0.9$ | $\hat{b} = 0.9$ | $\hat{b} = 0.9$ | $\hat{b} = 0.9$ | $\hat{b} = 2/3$ |
| Model | Parameter $\check{b}$ | $\check{b} = 0.55$ | $\check{b} = 0.60$ | $\check{b} = 0.65$ | $\check{b} = 2/3$ | $\check{b} = 0.70$ | $\check{b} = 0.75$ | $\check{b} = 2/3$ |
|---|---|---|---|---|---|---|---|---|
| MNIST | Non-targeted | 17121 | 17481 | 17151 | **17016** | 17440 | 17447 | 17091 |
| CNN | Targeted | 24750 | 24032 | **23863** | 24106 | 24960 | 24389 | 24166 |
| CIFAR10 | Non-targeted | 2908.0 | 2808.7 | 2786.1 | **2771.8** | 2833.1 | 2887.9 | 2791.0 |
| CNN | Targeted | 9954.0 | 9271.4 | 8827.8 | 9027.1 | **8804.8** | 9157.8 | 9046.2 |
| CIFAR100 | Non-targeted | 11359 | 10860 | 9926.5 | **9607.3** | 10102 | 11970 | 9621.3 |
| ViT | Targeted | 90993 | 94329 | 85023 | **83471** | 100429 | 90909 | 87365 |
| ImageNet | Non-targeted | 16913 | 16037 | 17005 | 16072 | **15366** | 16250 | 16182 |
| VGG19 | Targeted | 201410 | 198460 | 196677 | **193161** | 193700 | 198519 | 198932 |

*Table 2.* AUC of TtBA under different parameters $(\hat{b}, \check{b})$.

non-targeted and targeted attacks. These models have been introduced in Subsection 5.1. The results are presented in Table 2 and Table 3, with the best values highlighted in bold.

Our experiment results clearly indicate that the parameter settings $(\hat{b} = 0.9, \check{b} = 2/3)$ and $(\hat{k} = 0.2, \check{k} = 0.1)$ can achieve the best performance in five out of eight experiments. Even when they are not the top-performing configurations, their results remain very close to the best values.

Furthermore, the parameter setting $\hat{b} = 2/3$ and $\check{b} = 2/3$, as shown in Table 3, corresponds to TtBA without the local optima escape mechanism. Under this setting, TtBA updates the weight parameter as $k^{i+1} = 2/3 k_{\text{bridge}}^i$ in lines 18-24 of Algorithm 1, completely ignoring the curvature of the decision boundary. Our experiments show that this configuration achieved consistently lower performance compared to setting $\hat{b} = 0.9$ and $\check{b} = 2/3$ in Algorithm 1. This result highlights the importance of using our proposed mechanism to escape local optima.

We also conduct additional sensitivity analysis on robust models. As demonstrated in Table 4, TtBA can effectively attack robust models after adjusting its hyperparameters. Specifically, we modify the setting of $k = \check{b} \cdot k_{\text{bridge}}$ by varying the default value of $\check{b} = 2/3$ across $\{0.55, 0.575, 0.60, 0.625, 0.65, 2/3, 0.70\}$, and evaluate the AUC of two WRN models on the CIFAR-100 and TinyImageNet datasets. The results demonstrate that, for robust models, the setting $\check{b} = 0.625$ achieves the best performance in 3 out of 4 experiments, clearly surpassing the $\check{b} = 2/3$ setting. This difference likely

| Dataset | Parameter $\hat{k}$ | $\hat{k} = 0.15$ | $\hat{k} = 0.20$ | $\hat{k} = 0.15$ | $\hat{k} = 0.20$ | $\hat{k} = 0.25$ | $\hat{k} = 0.20$ | $\hat{k} = 0.25$ |
|---|---|---|---|---|---|---|---|---|
| Model | Parameter $\check{k}$ | $\check{k} = 0.05$ | $\check{k} = 0.05$ | $\check{k} = 0.10$ | $\check{k} = 0.10$ | $\check{k} = 0.10$ | $\check{k} = 0.15$ | $\check{k} = 0.15$ |
| MNIST | Non-targeted | 16972 | 17005 | 16954 | 17016 | **16946** | 17051 | 17008 |
| CNN | Targeted | **23745** | 24262 | 23972 | 24106 | 24389 | 24272 | 24601 |
| CIFAR10 | Non-targeted | 2777.0 | 2786.5 | 2781.2 | **2771.8** | 2787.2 | 2779.7 | 2774.0 |
| CNN | Targeted | 9340.9 | 9307.6 | 9126.3 | **9027.1** | 9221.4 | 9471.9 | 9350.8 |
| CIFAR100 | Non-targeted | 9711.4 | 9703.6 | 9698.2 | **9607.3** | 9614.4 | 9625.9 | 9688.1 |
| ViT | Targeted | 87940 | 88777 | 96075 | **83471** | 100613 | 106421 | 90233.9 |
| ImageNet | Non-targeted | 16199 | 16175 | 16120 | **16072** | 16167 | 16089 | 16168 |
| VGG19 | Targeted | 198385 | 192804 | 216609 | 193161 | **191641** | 216601 | 192685 |

*Table 3.* AUC of TtBA under different parameters $(\hat{k}, \check{k})$.

| Dataset | Parameter $\hat{b}$ | $\hat{b} = 0.9$ | $\hat{b} = 0.9$ | $\hat{b} = 0.9$ | $\hat{b} = 0.9$ | $\hat{b} = 0.9$ | $\hat{b} = 0.9$ | $\hat{b} = 0.9$ |
|---|---|---|---|---|---|---|---|---|
| Model | Parameter $\check{b}$ | $\check{b} = 0.55$ | $\check{b} = 0.575$ | $\check{b} = 0.60$ | $\check{b} = 0.625$ | $\check{b} = 0.65$ | $\check{b} = 2/3$ | $\check{b} = 0.70$ |
| CIFAR100 | Non-targeted | 8763.6 | 8790.2 | 8657.4 | **8605.4** | 8681.8 | 8784.6 | 8816.2 |
| WRN | Targeted | 22786 | 22288 | 21977 | **20799** | 22806 | 22973 | 23172 |
| TinyImageNet | Non-targeted | 31864 | 31230 | 30898 | **29437** | 29978 | 30026 | 30569 |
| WRN | Targeted | 121442 | 120874 | **115260** | 115681 | 116891 | 116976 | 117997 |

*Table 4.* AUC of TtBA under different parameters $(\hat{b}, \check{b})$ against robust trained models.

arises because robust models can effectively conceal gradient information, causing normal vector estimation to become less reliable. Consequently, assigning a smaller weight to the normal vector can enhance the effectiveness of perturbation optimization.

# H. Appendix Experiment on Attack Success Rate

The reduction in query complexity under the same perturbation budget is shown in Table 5. We evaluate the performance of TtBA on the ImageNet dataset across four models: VGG-19, ResNet-50, Inception-V3, and ViT-B32. Following the setup of CGBA, we set the query budget to 10,000 and the maximum $\ell_2$ perturbation strength to $\epsilon = 2.5$. We then randomly choose 500 images from ImageNet and compare the Attack Success Rate (ASR) and the average (median) number of queries required for a successful attack. The results are presented below.

| | Model | VGG | ResNet | Inception | ViT |
|---|---|---|---|---|---|
| Attack | | -19 | -50 | -V3 | -B32 |
| HSJA | Query | **2051.1(1071.8)** | **1833.8(1209.5)** | **2851.1(2080.1)** | **1873.9(947.5)** |
| | ASR | 61.0% | 38.8% | 57.2% | 59.6% |
| CGBA | Query | 2500.9(1528.5) | 3450.7(2679.0) | 3169.3(2363.0) | 2447.8(1797.0) |
| | ASR | 88.2% | 52.0% | 74.4% | 79.6% |
| TtBA | Query | 2350.8(1481.0) | 3546.6(2754.0) | 3098.8(2175.0) | 2384.4(1781.5) |
| | ASR | **93.2%** | **61.8%** | **80.0%** | **80.4%** |

*Table 5.* Comparison of decision-based untargeted attacks on the ImageNet dataset with 10,000 query budgets and maximum $\ell_2$ perturbation strengths $\epsilon = 2.5$.

The results show that TtBA achieves the highest attack success rate (ASR) across all four models. Regarding the number of queries, HSJA has the lowest average (median) number of queries, but this is due to its much lower ASR compared to CGBA and TtBA. As is well-known, some images contain robust features that require more queries for a successful attack. TtBA, with has significantly higher ASR, is able to successfully attack these robust images, thus requiring more queries on average. Meanwhile, with a similar ASR, TtBA outperforms CGBA in terms of average (median) queries. On ResNet-50, TtBA also achieves significantly higher ASR (61.8%) compared to CGBA (52.0%).

