# OpenReview forum: "TtBA: Two-third Bridge Approach for Decision-Based Adversarial Attack"
_ICML.cc/2025/Conference — ICML 2025 poster_

### Official Review · Reviewer_EhyV · 2025-03-07

**Overall Recommendation:** 4

**Summary:**

The paper proposes a novel decision-based black box attack against image classifiers. The attack is called TtBA and it is based upon exploiting the geometry of the decision boundary. It introduces a notion of the $k_{bridge}$ metric and discusses how it helps in constructing an efficient adversarial example. The attack is demonstrated to outperform the existing methods on a wide range of datasets and models (undefended and defended) in the untargeted and targeted attack settings.

**Claims And Evidence:**

The claims made in the paper are supported by empirical observations, formal analysis (Appendix) and extensive experimental results (Section 5 and Appendix).

**Essential References Not Discussed:**

There appear to be no missing essential references.

**Experimental Designs Or Analyses:**

It is not entirely clear whether the formulation of a targeted attack is reasonable (see the Methods And Evaluation Criteria Section for details).

Other than that there appear to be no issues.

Adversarial example definition in the line 199 doesn't contain clipping the resulting image to [0, 1], probably to do saving some space in the text. But in the code provided along with the submission it was checked that the clamping is happening.

**Methods And Evaluation Criteria:**

In the line 169 targeted attack is formulated via a target image $x_{target}$ with its corresponding label $f(x_{target})$ rather than a more typical formulation via a target label $y_{target}$ i. e. achieving $f(x)=y_target$. It is not clear whether evaluating attack efficiency in the targeted setting with this formulation is reasonable.

Other than that the evaluation methods and criteria seem to make sense for this problem.

**Other Comments Or Suggestions:**

No further comments.

**Other Strengths And Weaknesses:**

Strengths

1. A novel black-box attack outperforming state of the art, as demonstrated in extensive experiments.
2. Studying decision-based attacks is important because they are more practical than other types of black box attacks and can pose a significant threat for safety-critical machine learning applications.

Weaknesses

1. Figure 1 is a bit overloaded with notation. It would be good to simplify it.

**Questions For Authors:**

1. Have you observed any difference when analysing the curve of the decision boundary (e. g. Figure3) for undefended and adversarially trained models?

2. Does sensitivity analysis in the Tables 2 and 3 include adversarially robust models? Would you expect some other hyperparameters to work better for them?

**Relation To Broader Scientific Literature:**

The paper relates to the previous analysis of the decision boundary curvature and norm vector-based attacks.

**Theoretical Claims:**

There appear to be no proofs in the main part. The derivation in the Appendix were not carefully checked.

---

> ### Author Rebuttal · Authors · 2025-03-31
>
> We sincerely appreciate the reviewer's meticulous evaluation and valuable comments, which have greatly helped improve our manuscript.
>
> 1. The **problem formulation is reasonable** for two key reasons. **First**, in hard-label black-box attacks (e.g., HSJA, TA, CGBA), where only the model’s output label is available and gradient information is inaccessible, generating an initial adversarial example for a target label in targeted attacks is nearly impossible. As a result, these methods require a target image $ y_{\text{target}}$ as the initial adversarial example. We will explicitly highlight this methodological distinction in our revision to prevent potential confusion for readers. **Second**, HSJA, TA, and CGBA all evaluate attack efficiency under this targeted setting in their experiments. To ensure a fair comparison, we adopt the same setting in our experiments.
>
> 2. We agree that the adversarial example definition should explicitly include **clipping to [0,1]**. This was properly implemented in our code but inadvertently omitted from the text. We will add this clarification in the revision. We sincerely thank the reviewer for carefully checking our code.
>
> 3. Following your valuable advice, we will carefully **simplify Figure 1** in the revised paper.
>
> 4. Agreeing with the reviewer, in Figure 2 and 3, we observe that **models without defense** (i.e., undefended models) usually have decision boundaries with low curvature and large $k_{\text{bridge}}$, while robust models normally have high curvature and small $k_{\text{bridge}}$. The table below shows the relationship between average $k_{\text{bridge}}$ and average $\ell_2$ distortion for successful attacks on CIFAR100 and CIFAR10 datasets across undefended ViT, CNN, and defended WRN models. It demonstrates that **robust WRN models generally have higher average $\ell_2$ distortion and lower $k_{\text{bridge}}$**, suggesting that adversarially trained models are associated with high decision boundary curvature.
>
> | Model (Dataset) | ViT (CIFAR100) | WRN (CIFAR100) | CNN (CIFAR10) | WRN (CIFAR10) |
> |----------------|----------------|----------------|---------------|---------------|
> | AVG ℓ₂         | 0.779          | 0.991          | 0.180         | 1.198         |
> | AVG K_bridge   | 0.370         | 0.350          | 0.362         | 0.341         |
>
> 5. The sensitivity analysis in Tables 2 and 3 does not include adversarially robust models. **Here we add a sensitivity analysis on robust models**. By adjusting the hyperparameters of TtBA, it can certainly yield better performance for robust models, as demonstrated in the following table. Specifically, we modify the setting of $  k = \check{b} \cdot k_\text{bridge} $ by varying the default value of $  \check{b} = 2/3 $ across {0.55, 0.575, 0.60, 0.625, 0.65, 2/3, 0.70}, and evaluate the AUC of two WRN models on the CIFAR-100 and TinyImageNet datasets. The results demonstrate that, for robust models, the setting $  \check{b} = 0.625 $ achieves the best performance in 3 out of 4 experiments, clearly surpassing the $\check{b} = 2/3$ setting. This difference likely arises because robust models can effectively conceal gradient information, causing normal vector estimation to become less reliable. Consequently, assigning a smaller weight to the normal vector can enhance the effectiveness of perturbation optimization.
>
> | Dataset (model)      | Attack Type  | b̌=0.55 | b̌=0.575 | b̌=0.60 | b̌=0.625 | b̌=0.65 | b̌=2/3 | b̌=0.70 |
> |-------------|-------------|--------|---------|--------|---------|--------|-------|--------|
> | **CIFAR100** (WRN) | Non-targeted | 8763.6 | 8790.2 | 8657.4 | **8605.4** | 8681.8 | 8784.6 | 8816.2 |
> |              | Targeted    | 22786  | 22288   | 21977  | **20799** | 22806  | 22973 | 23172  |
> | **TinyImageNet** (WRN) | Non-targeted | 31864 | 31230 | 30898 | **29437** | 29978 | 30026 | 30569 |
> |              | Targeted    | 121442 | 120874 | **115260** | 115681 | 116891 | 116976 | 117997 |

---

> > ### Comment · Reviewer_EhyV · 2025-04-02
> >
> > I would like to thank the authors for addressing the points raised in my review. I have no further questions and I am keeping my score.

---

> > > ### Author Response · Authors · 2025-04-06
> > >
> > > Thank you again for your careful review and for checking our code. Your comment reminded us to clarify the targeted attack formulation and the definition of adversarial examples, which is crucial for eliminating misunderstandings and improving our work.

---

### Official Review · Reviewer_i6yX · 2025-03-10

**Overall Recommendation:** 3

**Summary:**

This paper introduces a decision-based black-box adversarial attack, termed Two-third Bridge Approach---TtBA, that focuses on optimizing perturbation directions for attack queries by leveraging normal vectors and the *bridge* direction, to relieve query complexity. This *bridge* direction is a weighted combo of the current perturbation direction and its normal vector, where the weight parameter is k. Through empirical evaluations, the authors show $k=\frac{2}{3}k_{\text{bridge}}$ offers the optimal directional alignment. With validations on various datasets, TtBA shows improved performances over existing non-targeted and targeted attack methods.

**Claims And Evidence:**

The claims made in the submission are partially supported by empirical evidence. The major concern of the reviewer is that, does the hypothesis that the decision boundary of DNNs is smooth and locally concave still hold for robust models? Only a robust WideResNet was studied in the experiments. The reviewer finds this to be slightly lacking. Are other adversarial defenses such as input transformation-based ones, and adversarial training techniques tailored for ViTs, etc., relevant under this context?

**Essential References Not Discussed:**

A few major decision-based attacks introduced in [1-2] are missing in the current experimental comparisons. The authors should discuss the relationship of TtBA to these works and/or explain why they were not included.

[1] Wan, Jie, Jianhao Fu, Lijin Wang, and Ziqi Yang. “BounceAttack: A Query-Efficient Decision-Based Adversarial Attack by Bouncing into the Wild.” In 2024 IEEE Symposium on Security and Privacy (SP), 1270–86, 2024. https://doi.org/10.1109/SP54263.2024.00068.

[2] Park, Jeonghwan, Paul Miller, and Niall McLaughlin. “Hard-Label Based Small Query Black-Box Adversarial Attack.” In 2024 IEEE/CVF Winter Conference on Applications of Computer Vision (WACV), 3974–83. Waikoloa, HI, USA: IEEE, 2024. https://doi.org/10.1109/WACV57701.2024.00394.

**Experimental Designs Or Analyses:**

Experimental designs are valid and supportive of the effectiveness of this proposed method.

**Methods And Evaluation Criteria:**

The evaluation setups are reasonable and supportive of this work.

**Other Comments Or Suggestions:**

N/A

**Other Strengths And Weaknesses:**

**Strengths:**
- The authors are to be congratulated with the extensive experiments on all 5 datasets with various common model architectures that include CNNs and ViTs.
- One of the major concerns, i.e., why 2/3 is the optimal parameter, has been explained in detail and empirically validated (in Figure 4 and Appendix G).

**Weaknesses**:
- Apart from the concerns mentioned above, the reviewer find that the experiments are quite focused on comparing perturbation size ($\ell_2$ distance) under the same query budget. The reviewer was kind of expecting that the work reports on the inverse, that is, the query complexity reduction under the same perturbation budget?

**Questions For Authors:**

N/A

**Relation To Broader Scientific Literature:**

The paper makes significant contributions to the broader scientific literature on black-box adversarial attacks, where research on decision-based hard label attacks is essential in theoretical and practical advancements.

**Theoretical Claims:**

The reviewer briefly went through the proofs in the Appendix and find that they are supportive of the claims made, but the reviewer did not check the correctness of the proofs.

---

> ### Author Rebuttal · Authors · 2025-03-31
>
> Thank you for your comments.
>
> 1. SOTA studies, including HSJA, TA, CGBA, strongly support the hypothesis that **the decision boundary of DNNs remains smooth and locally concave even for many robustly trained models**. This is because the robust training process does not interfere with normal vector estimation. We conduct additional experiments using the **Towards Robust Vision Transformer (RVT)** defense from [1], which enhances ViT robustness with position-aware attention scaling and patch-wise augmentation. The corresponding decision boundary is plotted in a figure at [decision boundary of robustly trained models](https://anonymous.4open.science/r/TtBA-6ECF/
> DecisionBoundaryofRobustViT.pdf). In the figure, the decision boundary remains smooth and locally concave.
>
>     However, for **input transformation-based defenses such as RandResizePad** in [2], the normal vector estimation process can be disrupted. This causes the decision boundary to lose its smoothness, which is shown at [abnormal decision boundary of RandResizePad](https://anonymous.4open.science/r/TtBA-6ECF/DecisionBoundaryofRandResizePad.pdf). It is important to note that **this issue is not specific to our approach**. All normal vector-based attacks, including HSJA, TA, and CGBA, encounter similar challenges and currently lack effective solutions. Since addressing these specific challenges is beyond the primary scope of our study, we did not include experiments on transformation-based defenses in our evaluation.
>
>     To strengthen our revised paper, we will expand the literature review to discuss attack methods specifically designed for input transformation-based defenses and highlight their differences from normal vector-based attacks. Additionally, we will update our future work section to explore potential extensions of our method to handle such defenses effectively.
>
> 2. We thank the reviewer for pointing out important attack methods such as BounceAttack and SQBA, which we have now **incorporated in the literature review section**.
>
> - While **BounceAttack** improves upon HSJA by using orthogonal gradient components and introduces momentum/smooth search mechanisms, it does not address the local optima problem caused by high-curvature decision boundaries, which is the core focus of our work. We would gladly compare with BounceAttack, but the official code is currently inaccessible, preventing us from presenting detailed results.
>
> - **SQBA** uses pre-trained surrogate models for gradient estimation and relies on access to the target model's training dataset. In contrast, our decision-based attacks assume no access to the training dataset. Therefore, a direct experimental comparison with SQBA will not be included. Thank you for your understanding!
>
> 3. **The reduction in query complexity under the same perturbation budget** is shown below. Following the setup of CGBA, we set the query budget to 10,000 and the maximum $ \ell_2$ perturbation strength to $ \epsilon = 2.5$. We then randomly choose 500 images from ImageNet and compare the Attack Success Rate (ASR) and the average (median) queries.
>
> | Attack | Model  | VGG-19            | ResNet-50        | Inception-V3      | ViT-B32          |
> |--------|--------|-------------------|------------------|-------------------|------------------|
> | HSJA   | Query  | **2051.1(1071.8)** | **1833.8(1209.5)** | **2851.1(2080.1)** | **1873.9(947.5)** |
> |        | ASR    | 61.0%             | 38.8%            | 57.2%             | 59.6%            |
> | CGBA   | Query  | 2500.9(1528.5)     | 3450.7(2679.0)   | 3169.3(2363.0)     | 2447.8(1797.0)   |
> |        | ASR    | 88.2%             | 52.0%            | 74.4%             | 79.6%            |
> | TtBA   | Query  | 2350.8(1481.0)     | 3546.6(2754.0)   | 3098.8(2175.0)     | 2384.4(1781.5)   |
> |        | ASR    | **93.2%**         | **61.8%**        | **80.0%**         | **80.4%**        |
>
> - The results show that **TtBA achieves the highest ASR** across all models. HSJA has the lowest average (median) number of queries, but this is due to its much lower ASR. As is well-known, some images contain robust features that require more queries to attack. TtBA, with significantly higher ASR, is able to successfully attack these robust images, thus requiring more queries on average. Meanwhile, with a similar ASR, TtBA outperforms CGBA in terms of average (median) queries. On ResNet-50, TtBA also achieves significantly higher ASR (61.8\%) compared to CGBA (52.0\%). We will include these results in our revised paper.
>
> **We believe we have satisfactorily addressed all the concerns raised in our rebuttal. If the reviewer agrees, would you please kindly consider adjusting your rating?**
>
> [1] Mao, Xiaofeng, et al. "Towards robust vision transformer." In Proceedings of the IEEE/CVF conference on Computer Vision and Pattern Recognition. 2022.
>
> [2] Xie, C., Wang, J., et al. "Mitigating adversarial effects through randomization." In arXiv preprint arXiv:1711.01991.

---

> > ### Comment · Reviewer_i6yX · 2025-04-07
> >
> > Thanks for the clarifications. I feel most of my concerns are resolved. I will be keeping my score of 3: Weak accept.

---

> > > ### Author Response · Authors · 2025-04-07
> > >
> > > Thank you very much for your insightful and constructive comments and for providing essential references. Your suggestions have significantly enhanced our work.

---

### Official Review · Reviewer_1Mse · 2025-03-12

**Overall Recommendation:** 3

**Summary:**

The manuscript introduces an innovative bridge direction to optimize the adversarial perturbation by linearly combining the current unit perturbation direction with its unit normal vector. Via experiment observation, k= 2/3 k_{bridge} can yield a near-optimal perturbation direction. Besides, the paper designs a simple and effective approach to detect and escape the local optima, making the proposed method better than the SOTA.

**Claims And Evidence:**

The novelty claims and theoretical derivations are reasonable.

**Essential References Not Discussed:**

Not enough.

**Experimental Designs Or Analyses:**

The experiments and results are convincing.

**Methods And Evaluation Criteria:**

The proposed method and the used criteria, including evaluation datasets, are common and representative.

**Other Comments Or Suggestions:**

The paper includes certain novelty components, but seems to be not enough to get the bar of the ICML.

**Other Strengths And Weaknesses:**

Strongs: For targeted attacks, narrow adversarial regions lead to being more easily trapped in local optima.
Weakness: Why d_{bridge}^{i} can be ensured to have identical decision bourndary as \hat{d}^{i}, as shown in Figure 1.

**Questions For Authors:**

In total, the proposed method is only an improvement of existing techniques, mainly based on HSJA, TA, qFool, GeoDA, QEBA, and CGBA. No brand-new insight is found to contribute to AI security fields. Finding a k=2/3k_{bridge} by experiments matched with some theoretical verification, designing an escape scheme to skip the local optima, and so on are not too challenging and very innovative. The performance improvement is not very significant, some are even below the sota results.

**Relation To Broader Scientific Literature:**

The optimization strategy may be somewhat insightful to other fields.

**Theoretical Claims:**

I check the theoretical claims and the corresponding proofs are correct.

---

> ### Author Rebuttal · Authors · 2025-03-31
>
> Thank you for your valuable comments.
>
> 1. We **perform a binary search** of $k = k_\text{bridge}^{i} \in (0,1]$ to identify $d_k = d_\text{bridge}^{i}$ which **have identical decision boundary** as $\hat{d}^{i}$.
>
> - According to Figure 1, when $k$ is very small, direction $d_k$ approaches $\hat{d}^{i}$ and its decision boundary of $d_k$ is smaller than $\hat{d}^{i}$.
>
> - When $k=1$, $d_k = \hat{N}^i$, its decision boundary is significantly larger than $\hat{d}^{i}$. **By the intermediate value theorem, there must exist $k \in (0,1]$ such that $d_k$ yields the same decision boundary as $\hat{d}^{i}$**. We will clarify the above in the revised paper.
>
> 2. Thank you for raising concern regarding the "brand-new insight". **Our primary innovation** lies in identifying and rigorously analyzing the previously unknown relationship between decision boundary curvature and the optimization of adversarial perturbations. Existing SOTA decision based methods (HSJA, TA, QEBA, and CGBA) have largely overlooked how boundary curvature influences the occurrence of local optima that seriously impacts optimization efficiency and effectiveness.
>
>     In contrast, our work introduces a **novel and practical curvature metric**, $ k_\text{bridge}$, which provides the first systematic means to quantify and interpret decision boundary geometry. This new understanding allows us to pinpoint precisely why and how adversarial attacks fail under certain geometric conditions, delivering useful insights previously missing from the literature.
>
>     Leveraging this discovery, we developed TtBA, a significantly **more effective and efficient** decision-based attack method. Further, we introduced a **robust mechanism specifically designed to detect and escape local optima** induced by boundary curvature, directly addressing an important limitation unexplored by previous studies. In summary, rather than merely improving upon prior methods, our research **introduces brand-new conceptual understanding and practical tools** with significant contributions to the AI security field. We will further clarify the above discussion in the revised paper.
>
> 3. We acknowledge that the proposed techniques may **appear conceptually simple** at first glance. However, identifying critical yet overlooked issues, including the presence of local optima due to high-curvature decision boundaries, is far from trivial. Developing practical, intuitive, and effective solutions to address such issues further highlights the strength and novelty of our contributions. Hence, **the simplicity of our solution does not diminish its novelty or importance**; rather, it underscores the clarity and practical value of our research.
>
>     **The novelty of our technical contributions** lies precisely in uncovering significant issues that have not received sufficient attention in existing literature. Despite extensive research, prior SOTA decision based methods have largely ignored how boundary curvature leads to local optimization traps that severely hinder adversarial attacks. Our research uniquely identifies this critical gap and proposes robust, intuitive, and demonstrably effective methods to address it.
>
>     Therefore, while the proposed solutions might seem intuitive after being introduced, we argue that **recognizing and formulating these specific problems** and subsequently **developing simple yet powerful techniques** constitute substantial and novel contributions to the AI security community. We will further clarify the above discussion in the revised paper.
>
> 4. **Our evaluation is rigorous**, covering extensive experiments across five datasets and seven distinct model architectures, representing a **comprehensive and highly challenging benchmark**. It is noteworthy that consistently surpassing SOTA performance across all tests is exceptionally difficult, which is a challenge similarly faced by recent leading methods such as HSJA, TA, and CGBA. Despite this inherent difficulty, our method achieves substantial performance improvements. Specifically in Table 1, TtBA clearly outperforms existing SOTA methods in 103 out of 108 experimented scenarios, with few remaining cases closely matching the best performance.
>
>     Furthermore, for robust models evaluated in Figure 5, **we intentionally refrained from fine-tuning parameters** to rigorously test our method's robustness and generalization capability. Even under this conservative setting, we demonstrated superior performance in 37 out of 40 cases. Additional targeted parameter tuning can further enhance the effectiveness of TtBA. However, we deliberately emphasized our method's strong general performance and broad applicability across diverse settings, thereby reinforcing the substantial practical value and robustness of our contributions.
>
> **We believe we have satisfactorily addressed all the concerns raised in our rebuttal. If the reviewer agrees, would you please kindly consider adjusting your rating?**

---

> > ### Comment · Reviewer_1Mse · 2025-04-02
> >
> > Thank the authors for the careful feedback. After reading the rebuttals of the authors, I think most of my concerns are addressed. I will be willing to raise my rating.

---

> > > ### Author Response · Authors · 2025-04-06
> > >
> > > Thank you very much for your positive and encouraging review. We sincerely appreciate your valuable comments and constructive suggestions, which have helped improve our work.

---

### Official Review · Reviewer_UW3F · 2025-03-12

**Overall Recommendation:** 3

**Summary:**

The paper proposes the TtBA method for decision-based black-box adversarial attacks.
It introduces a new bridge direction, a weighted combination of the current direction and its normal vector, controlled by a weight parameter $k$.
Experiments on multiple datasets and models show that TtBA outperforms state-of-the-art methods in both targeted and non-targeted attacks.

**Claims And Evidence:**

Yes

**Essential References Not Discussed:**

n/a

**Experimental Designs Or Analyses:**

Yes

**Methods And Evaluation Criteria:**

Yes

**Other Comments Or Suggestions:**

n/a

**Other Strengths And Weaknesses:**

## Strength

- The proposed method improves the performance of decision-based adversarial attacks.

- The paper is well written.

## Weakness

- The contribution of this paper is a little weak, from my perspective.

- Some settings are somewhat empirical and without rational explanation. For example,  $k = 2/3k^i_{bridge}$.

**Questions For Authors:**

n/a

**Relation To Broader Scientific Literature:**

n/a

**Theoretical Claims:**

Yes

---

> ### Author Rebuttal · Authors · 2025-03-31
>
> 1. Thank you for raising concerns regarding **the strength of our contributions**. We introduce a fundamentally new and practically valuable metric, $k_\text{bridge}$, specifically designed to quantify decision boundary curvature, a critical but previously unexplored factor in adversarial attacks. This metric reveals vital insights into how geometric properties of decision boundaries directly affect the effectiveness of decision-based attacks. Existing SOTA decision based attack methods such as HSJA, TA, qFool, GeoDA, QEBA, and CGBA **overlook the critical issue of local optima caused by high-curvature boundaries**, significantly reducing their attack effectiveness. Addressing this gap, we make three substantial contributions:
>
>     (1) We propose $k_\text{bridge}$, the first quantitative metric in literature, to **rigorously measure boundary curvature**, enabling systematic analysis and deeper understanding of adversarial optimization.
>
>     (2) Using insights from $ k_\text{bridge}$, we uncover a **previously unidentified linear relationship between boundary curvature and optimal perturbation directions**. Leveraging this discovery, we develop the TtBA method for highly effective decision-based black-box attack.
>
>     (3) We **identify a low attack efficiency problem caused by high boundary curvature** and propose a robust mechanism to detect and escape them, significantly enhancing optimization efficiency and attack success rates.
>
>     Our extensive experiments across multiple widely-used datasets and models clearly demonstrate the substantial practical impact of our contributions, representing a **significant advancement in adversarial machine learning**. We will further clarify the above discussion in the revised paper.
>
> 2. While the setting $k=2/3k_\text{bridge}^{i}$ is empirically motivated, it is also **supported by an extensive sensitivity analysis in Appendix G**. Specifically, we analyze the sensitivity of our method to different settings of $k$ by varying the default value of $2/3$ across {0.55, 0.60, 0.65, 0.70, 0.75} in Table 2, and similarly adjusting other parameters in Table 3. Our results reveal two key findings: first, the current configuration achieves the best performance in 10 out of 16 experimental scenarios; second, alternative parameter values maintain comparable effectiveness. These results demonstrate that, while $k=2/3$ represents the most effective choice, TtBA's performance remains robust to parameter variations, ensuring methodological reliability across different configurations.
>
> **We believe we have satisfactorily addressed all the concerns raised in our rebuttal. If the reviewer agrees, would you please kindly consider adjusting your rating?**

---

### Decision · Program_Chairs · 2025-05-01

**Decision:**

Accept (poster)

**Comment:**

Reviewers identified some relatively minor issues regarding additional related work, clarification of the threat model and its access to data, and other clarifying questions. These were all addressed during the response period. Now, it is not entirely obvious why this adjusted threat model, which deviates from prior work, is of utmost importance to the broader ML community. Beyond being merely a methodological difference, the value of this work would be a lot more valuable if connected to more real-world challenges or considerations reflecting actual use-cases of the threat model. As a result, while the work is good and could be interesting, its general value is not totally clear even after reading the discussion, leading to my recommendation of weak accept.